# The Impacts of Pilates and Yoga on Health-Promoting Behaviors and Subjective Health Status

**DOI:** 10.3390/ijerph18073802

**Published:** 2021-04-06

**Authors:** Eun-Ju Lim, Eun-Jung Hyun

**Affiliations:** 1Department of Exercise Rehabilitation & Welfare, Gachon University, Incheon 21936, Korea; ejblue31@naver.com; 2Department of Business, Hongik University, Seoul 04066, Korea

**Keywords:** Pilates, yoga, health-promoting lifestyle profile, self-reported health status

## Abstract

This study investigates whether Pilates and yoga lead people to adopt generally health-promoting lifestyle elements and feel better about their physical and mental fitness. To this end, we designed an 8 week exercise program of Pilates and yoga reviewed by veteran practitioners and conducted an experimental study through which we collected the data from 90 volunteered adult subjects between ages 30 and 49 (mean age = 35.47), equally represented by women and men without previous experience with Pilates or yoga. In the 8 week long experiment, we assigned the subjects to three groups, where subjects in the two exercise groups regularly took part in either Pilates or yoga classes, and the control group participated in neither exercise classes. All participants completed two surveys, the Health-Promoting Lifestyle Profile (HPLP II) and the Health Self-Rating Scale (HSRS), before and after their assigned program. In our analysis of pre- and post-treatment differences across the three groups, we ran ANOVA, ANCOVA, and Sheffé test, implemented using SPSS PASW Statistics 18.00. Our results indicate that Pilates and yoga groups exhibited a higher engagement in health-promoting behaviors than the control group after the program. Subjective health status, measured with HSRS, also improved significantly among Pilates and yoga participants compared to those in the control group after the program. The supplementary analysis finds no significant gender-based difference in these impacts. Overall, our results confirm that Pilates and yoga help recruit health-promoting behaviors in participants and engender positive beliefs about their subjective health status, thereby setting a positive reinforcement cycle in motion. By providing clear evidence that the promotion of Pilates or yoga can serve as an effective intervention strategy that helps individuals change behaviors adverse to their health, this study offers practical implications for healthcare professionals and public health officials alike.

## 1. Introduction

Exercise is well documented as having various health benefits. Abundant evidence suggests that regular physical activity is associated with lower obesity rate and cardiovascular disease incidence, better sleep patterns and sexual function, and slower aging-related deterioration of the immune system [1,2]. Individuals who exercise routinely also tend to report better mood states than those who do not [1]. Indeed, exercise is generally accepted as an integral component of health-promoting behavior, defined as a broad set of lifestyle elements—including consuming nutritious foods, maintaining adequate sleep, minimizing stress, and staying away from health-detrimental habits such as smoking—positively related to better health. Most physicians subscribe to the idea that exercise helps mitigate the risk of disease, prevent premature death, and improve the overall quality of life. From a public health standpoint, exercise is one of the most cost-effective means by which public health goals can be attained [3]. The notion that exercise is essential for leading a healthy life, both physically and psychologically, is now indisputable.

Among many kinds of physical activity programs, it is noteworthy that Pilates and yoga have gained increasing popularity amongst the general public over the past two decades. Pilates and yoga are particularly appealing due to their direct benefits on physical wellbeing—including weight control and improved posture, flexibility, and cardiovascular function—that come with low risks of sports-related injuries [3]. According to an annual survey conducted by IDEA(International Dance Exercise Association) Fitness Programs and Equipment Survey in 2007, Pilates ranked sixth on the most frequently offered exercise programs, a vast improvement since 1999 [4]. In the same year, yoga also ranked 13th, although its position has undergone gradual declines from its peak in 2002. In annual Fitness Trends Surveys carried out by a United States (US)-based association of Sports Medicine, Pilates and yoga have been frequently listed as Worldwide Fitness Trends since 2008 [5].

Evidence of the direct health benefits of Pilates and yoga is growing. For example, some studies showed that regular engagement in Pilates is associated with a boost in functional autonomy, balance, flexibility, and muscle strength [6,7,8,9,10,11,12]. Other studies show that regular yoga participation helped individuals alleviate muscle-related pains, especially among adults with sedentary lifestyles or suffering from chronic illnesses [9,10].

Less known, however, is whether Pilates and yoga may also have secondary, indirect benefits. Adopting a broad set of lifestyle elements that are generally regarded as beneficial to one’s physical and psychological health and holding a positive belief about one’s health might be necessary for leading a healthy life. Some prior research hints at the possibility that one form of health-promoting behavior triggers people to adopt broadly health-promoting lifestyle elements (e.g., healthy eating, enriching interpersonal relationships, informed health-related decisions, work–life balance, and emphasis on spiritual growth), thereby setting in motion a positive reinforcement loop. For example, Küçük and Livanelioglu (2015) found that individuals starting a clinical Pilates program tend to hold a positive notion about the goodness of exercises and experience an improvement in their sociopsychological aspects of life [11]. Mustian’s research team (2013) found that yoga has a positive impact on the sleep quality among cancer survivors [13]. Galasso and colleagues (2020) found that aerobic and anaerobic exercise training was effective in reducing the tendency of binge eating [14].

Nonetheless, these studies come short of providing more comprehensive and clear evidence that Pilates leads to a positive change in the participant’s overall health-promoting behavior and self-perceived health status. Furthermore, Küçük and Livanelioglu (2015)’s study primarily targeted people taking a clinical Pilates program, excluding those who might regularly engage in Pilates for nonclinical purposes [11]. Moreover, Pilates was not covered in studies by Mustian et al. (2013) and Galasso et al. (2020). Given this, the current study seeks to fill this deficiency in prior work by directly investigating whether regularly engaging in an exercise program such as Pilates and yoga might positively affect participants’ adoption of health-promoting behaviors. Additionally, we examine the extent to which these outcomes shown in program participants are accompanied by an increase in self-perceived physical and mental wellbeing.

In the subsequent sections, we first provide a brief overview of the origins and principles of Pilates and yoga. Next, extending the literature on Pilates and yoga’s health benefits, we develop our hypotheses about the relationship between participation in Pilates and yoga on people’s health-promoting behavior and subjective perception of wellbeing. In the following sections, we provide a detailed description of our research methods and report our findings. In the final section, we discuss our work’s contributions and limitations, with some remarks on our study’s practical implications.

## 2. Research Background and Hypotheses

### 2.1. The Origins and Principles of Pilates and Yoga

Initially called “Contrology”, Pilates is a physical fitness system and exercise method developed by Joseph H. Pilates from Germany during World War I [15]. Pilates was initially put together to provide rehabilitation to prison-camp inmates during the war. When Joseph Pilates immigrated to the US at the close of World War I, he launched a long-term collaboration with dancers and celebrities interested in fitness, which led to many refinements in his earlier exercise methods. During this collaboration, his exercise methods were referred to as Pilates, named after the inventor, Joseph H. Pilates. With the development of a set of assistive equipment, including the Cadillac, Universal Reformer, Chair, Barrel, and Pilates, Pilates is now widely accepted as a useful exercise method for core strength and rehabilitation. The first-generation Pilates trainers, including Lolita San Miguel and Mary Bowen, have actively promoted this novel physical activity method to the general public. Pilates’ original principles lie in centering, concentration, control, precision, flow, and breathing [16]. Many Pilates instructors expand from these original principles when developing new applications for training. For example, Pilates Method Alliance (PMA) is a combination of “balanced muscle development”, “whole-body movement”, and “rhythm as flow”, each of which derives from the original set of Pilates outcomes [15]. The Balanced Body Pilates curriculum is primarily a mixture of relaxation and PMA’s main principles [17]. With these developments designed to emphasize core training, the Pilates method has ultimately achieved its aim to promote flexibility, endurance, circulation, strength, and body balance. Moreover, Pilates maintains good posture by strengthening lumbopelvic stability, developing muscle tone, protecting the back, and optimizing the spine’s alignment through balance.

Rooted in ancient India, yoga has a Sanskrit etymological origin with the meaning of “union”. More precisely, yoga refers to the union of mind with the universe’s divine. According to Selvalakshmi (2015), yoga purports to liberate a human being from body–mind duality conflicts in every living thing [18]. While yoga gained its early popularity in the Western world, it has gradually established its status as a joint exercise and meditation practice worldwide. Yoga emphasizes harmony among physical, mental, and spiritual elements. Various yoga methods have evolved from their original roots inspired by multiple elements from Hinduism, Buddhism, and Jainism. Among numerous derivative forms of yoga, Hatha yoga is one of the most popular types to date practiced outside India, with its characteristics most akin to physical exercise. Hatha yoga incorporates activities that combine body, mind, and breath. Iyengar yoga, put together by B. K. S. Iyengar, belongs to Hatha yoga that we focus on in this study [19]. The exercise methods in Iyengar yoga rely on “asana” (poses), in tandem with focused breathing (pranayama) and meditation, to improve flexibility, mobility, stability, and strength and to facilitate relaxation. Iyengar yoga consists of over 200 classical yoga asanas with 14 different pranayama types, ranging from basic to advanced. Students progress gradually from learning simple techniques to building up to more complicated ones.

### 2.2. The Impacts of Pilates and Yoga on Health-Promoting Behavior and Subjective Health Status 

Pilates aims to achieve full control of one’s behavior through mind–body coordination and fitness, as its original name “Contrology” implies. Similarly, yoga was developed and evolved to achieve the integration of mind, body, and spirit. The PMA guide highlights that Pilates is designed to release stress, aid fatigue recovery, facilitate spiritual rejuvenation, and heighten self-awareness and self-confidence [17]. Yoga practitioners also aim to achieve similar goals [19]. People who practice Pilates or yoga do so not just to increase the amount of their physical activity but also to restore a balance in their hectic everyday lives. So far, extant research supports some of these claims and intuitions [6,7,8,9,10,11,12].

Indeed, a growing body of evidence suggests that regularly practicing Pilates or yoga brings many benefits for the people and their physical, emotional, and psychological wellbeing. Pilates and yoga’s health-related benefits range from more tangible ones to more subtle ones. For example, Roh (2016) reported the positive impact of Pilates on positive physical self-description and subjective happiness among Pilates participants [20]. Bullo and colleagues (2015) also reported that Pilates is associated with improved mood states and quality of life among older adults [21]. Mokhtari, Nezakatalhossaini, and Esfarjani (2013) reported similar findings in investigating the health benefits among older adults [22]. Even among healthy individuals, evidence suggests that Pilates is, in general, instrumental in developing dynamic balance, flexibility, and muscle tone [23].

Many published studies also reported similar health benefits of yoga [24,25,26,27,28]. Selvalakshmi (2015) argued that yoga is an effective therapy in reducing physiological discomforts among postnatal care women [18]. Koo and Shim (2015) found that Zen yoga effectively alleviates sympathetic nerve activation and relieves psychological frustration [24]. Lim and Cheong’s study (2012) demonstrated that yoga classes, initiated as part of a community welfare program, proved helpful in reducing anxiety and depression among the lower-income bracket females [25]. In a similar vein, Kim and Kim (2012) reported that yoga and meditation programs effectively alleviated depression among middle-aged women and positively influenced their social role performance and self-confidence [26].

Despite the increasing popularity of Pilates and yoga among adults who desire to escape from busy and sedentary lifestyles, Pilates and yoga’s precise effects on a healthy balance restoration in life are less well known [27,28]. One way to address this question is to directly investigate the effects of Pilates and yoga on the participant’s adoption of health-promoting behaviors and subjective wellbeing. If we can establish this relationship, a readily available, low-risk exercise option like Pilates or yoga can be considered a cost-effective way to set such a positively reinforcing cycle toward a healthy life with some confidence. Thus, in an attempt to address this question, the current study examines whether Pilates and yoga participation triggers health-promoting behavior and helps increase their subject health status among the participants.

Health-promoting behaviors refer to the activities geared toward promoting fitness and reducing physical and mental damages. Subjective health status refers to a self-assessment or beliefs about one’s health quality. A core assumption underlying these concepts of health-promoting action and subjective health status is that people have a sufficient degree of control over their health-related behavior and that beliefs about their health have some influence on their health-related behavior. Accordingly, we test the following two main hypotheses in this study:

**Hypothesis** **1:**Pilates and yoga groups will show a higher increase in health-promoting behaviors after completing their Pilates and yoga program than the control group.

**Hypothesis** **2:**Pilates and yoga groups will show a higher increase in subjective health status after completing their Pilates and yoga program than the control group.

## 3. Research Method

To test our hypotheses, we designed an experiment where our treatment was Pilates and yoga program intervention, and the two dependent variables of our interest were (1) post-treatment changes in the participants’ self-reporting of engagement in health-promoting behaviors (Health-Promoting Lifestyle Profile (HPLP) II) and (2) post-treatment changes in the participants’ self-reporting of subjective health status (Health Self-Rating Scale (HSRS)). Our analytical strategy compares pretreatment and post-treatment changes across the three groups, including the Pilates group, the yoga group, and the control group (no exercise group).

In October 2017, the first author obtained approval for this experiment in which human subjects were involved from the Institutional Review Board (IRB) of the Korean National Sports University, with which the first author was then affiliated.

### 3.1. HPLP II and HSRS

The Health Promoting Lifestyle Profile (HPLP) is a widely used assessment tool first developed by Walker, Sechrist, and Pender (1987, 1995) [29,30]. In their early work, these researchers focused primarily on developing psychometric scales [29]. Later, the original HPLP was updated to reflect a comprehensive nature of a person’s behaviors that has ramifications for their physical and mental fitness and evolved as the HPLP II [31].

HPLP II uses the four-point Likert scale with higher scores indicating greater participation in health-promoting activities and consists of 52 items that can be divided into six component clusters—(1) health responsibility, (2) physical activity, (3) nutrition, (4) spiritual growth (or self-actualization), (5) interpersonal relations (or interpersonal support), and (6) stress management [29,30]. Health responsibility captures how a person pays attention to and takes good care of their health on the basis of health professionals’ guides. Physical activity is a measure of whether a person does workouts of varying intensity and engages in physical activity of varying kinds and levels. Nutrition captures the extent to which a person consumes appropriate quality and quality foods. Spiritual growth measures the degree to which a person nurtures inner and leads a meaningful life. Interpersonal relations reflect how a person cares about maintaining a meaningful relationship with others and can connect socially and emotionally. Stress management measures the extent to which a person can release their stress and keep stressors in control [30]. The HPLP II has been deployed extensively by researchers interested in the relationships among these subscales and the impacts of a specific physical activity [31,32,33,34,35,36].

The Health Self-Rating Scale (HSRS) is a comprehensive self-assessment of health quality. HSRS was first developed by Lawton et al. (1982) and has been widely adopted across numerous studies on subjective health status. The HSRS survey asks participants to rate their present health status compared to how it was 1 year prior and how it compares to their friends or colleagues [37,38].

This study adopted a local adaptation of the original HPLP II and the original HSRS. We relied on Korean translations of HPLP II and HSRS performed by Seo and Hah (2004) and Shin and Kim (2009), respectively [36,37,38].

### 3.2. Intervention Program 

Two Pilates and two yoga experts were recruited as instructors for participants to formulate our intervention strategy. Throughout this study, the first author consulted these experts and relevant outside authorities when developing a 4 week pilot program of Pilates and yoga and an 8 week main program to be used as a treatment (intervention) in the experiment. The instructors hired for this study were all fully certified with international licenses. Pilates instructors were certified Balanced Body instructors from the US with more than three years of Pilates class teaching experience. Yoga instructors were also similarly qualified.

The intervention implemented in this study consisted of two independent exercise programs, Pilates and yoga. We incorporated activities from the Balanced Body instructor manuals [17]. The Pilates program contained exercises focusing on core strengthening, lumbopelvic stability, and flexibility. All yoga activities were also derived from well-established sources [19]. Our yoga exercises were based on the master of Hatha yoga, yoga Dipika. Similarly, the yoga program contained stretching, flexibility, and strengthening exercises.

Table A1 and Table A2 (Appendix A) detail each exercise’s purpose in our Pilates and yoga program, with precautions given in the instructor manuals. These exercises had the primary purpose of developing or increasing certain parts of the bodily function. Pilates and yoga group participants took part in their respective class with our trained instructors over the entire 8 week program at the designated location. Pilates typically uses several essential pieces of equipment, such as a Reformer, Chair Barrels, and Trapeze Table, which constitutes the original Pilates apparatus. However, in this study, we decided to focus only on mat-based Pilates to ensure that exercise sessions in Pilates and yoga were similarly formed to minimize the possibility that equipment usage in Pilates might interfere with our experiment as an additional unaccounted variable.

Sessions of 1 hour duration were held three times a week for eight weeks at each designated location. Exercise parameters (FITT: Frequency, Intensity, Time, Type) were as follows (see Table A3 and Table A4 in the Appendix A for details):Exercise frequency: three times per week;Exercise intensity: beginning/intermediate (three sets of six repetitions);Exercise time: 50 min per session (1 hour session including warm-up and cool-down) for 9=8 weeks;Exercise type: Pilates (mat), yoga.

The Pilates and yoga programs were similar in duration (three 1 h sessions per week for 8 weeks) and technique (both employed mat exercises only). Moreover, the instructors were informed to be careful not to coach participants on the other health-promoting behaviors being assessed. The most distinctive difference between the Pilates and yoga programs was how the exercise programs were conducted. For example, Pilates generally focuses on effective movement and is more likely to utilize dynamic, resistance, and stability-related poses. Yoga focuses typically on strength and flexibility by achieving relaxation and uses more stretching and holding static postures. Pilates instruction also tends to use more verbal explanations, while yoga relies more on demonstration.

#### Pilot Test

In November 2017, a 4 week pilot test was started with nine people who participated in a 4 week exercise program. Our 4 week pilot study aimed to ensure the appropriate design for our main experiment to be followed. Our focus was to configure the exercise program’s adequate task complexity levels in advance of the full-scale experiment to last over 8 weeks.

The pilot test subjects were informed that they were participating in a study on a community health program and that their personal information would be kept confidential. Participants were randomly assigned to each of the three groups: a 4 week Pilates program, a 4 week yoga program, or a control group with no specified exercise program.

During the pilot study, we noticed that some of the exercises we initially included in the program were quite challenging and problematic for some people. For example, the rollover movement in Pilates turned out to be rather hard for those with no prior exposure to Pilates and yoga. It was crucial to ensure that the exercises included in each program were safe and secure, as well as suitable for beginning and intermediate-level students with no Pilates or yoga experience. Given that people usually have a weak core and unstable lumbopelvic stability, the rollover, inversion movements, and extreme stretching were removed to minimize any injury or accident during the sessions.

### 3.3. Participants

Our inclusion criteria were age 30–49 years, with no severe disease and an interest in yoga and Pilates. Our exclusion criteria were having prior experience with yoga and Pilates and currently participating regularly in exercises such as swimming and regular fitness workouts.

The Yoga Community and Korea Pilates Federation assisted recruitment of participants and instructors. In November 2017, the first author contacted and visited the national Pilates Federation and several local yoga communities across South Korea. In meetings with directors of these organizations, the first author explained the purpose of this study. The directors and organizations who agreed to participate in this study assisted in recruiting volunteer participants. Potential participants were identified through these contacts and inquired about their interest and willingness to participate in this study. A sufficient number of people (approximately 100) expressed their desire to participate in our research.

Two exercise groups of 30 individuals were randomly selected with a balanced gender ratio for each group. The Pilates and yoga group participated in an 8 week predesigned exercise program.

We also needed a control group who would not receive treatment (i.e., no exercise) but would take the two surveys following their instructor’s directions. The control group did not participate in any of the specified exercise programs to serve as a baseline comparison for the experiment. Individuals in the control group comprised those who expressed interest in taking a Pilates or yoga class but could not make it for scheduling conflicts. However, they agreed to complete the two surveys at the same 8 week interval as the Pilates and yoga groups. Throughout the 8 week study, we ensured that they did not engage in any other exercise programs during the study period.

#### 3.3.1. Prescreening

A total of 90 volunteers participated in this project. All participants took their first Pilates or yoga class or did not have prior experience with Pilates or yoga. We provided them with a privacy statement stipulating that their personal information would remain strictly confidential.

Before starting the exercise program designed, we screened the pretreatment survey responses of our volunteer participants regarding their characteristics and exercise status to ensure that they were similar in those aspects. Through the survey and onsite measurement equipment, we obtained basic demographic information (gender, age, education level, occupation) and anthropometric data (height, weight, waist circumference, hip circumference, body mass index (BMI)). We measured the participants’ BMI using In-Body equipment at the studio (e.g., InBody Dial W version 2.3.05, Korea).

Of high importance was to ask the participants whether they were involved in any other exercise programs. This question was necessary to mitigate against any systematic bias that their prior or recent exposure to exercise programs of any kind can introduce. Later, we checked whether there was any systematic difference between people currently practicing other exercises and those not engaging in any other exercise program. These pilot test participants did not take part in the main experiment.

In the survey conducted at the pretest, a subject was to answer the three statements about their exercise status: (1) I do exercise for health; (2) I do regular exercise; (3) I know a proper exercise program. To ensure the homogeneity of different exercise status among the three groups, we ran a chi-square test to see whether there were differences between the three groups in exercise status.

We ran the homogeneity test to see if participants were equivalent. In particular, it was critical to check their exercise status because a participant’s exercise status can serve as a potential source of interference in our experiment due to its expected influence on HPLP II and HSRS. People who exercise for health and know of a proper exercise method turned out to be concentrated in the control group at a statistically meaningful level (X^2^ = 10.075, *p* < 0.01, X^2^ = 8.285, *p* < 0.05). This means that people who exercise for health and know of a proper exercise method were not randomly assigned to each group. Accordingly, the first author undertook an interview with those who answered positively to these questions to find out more about their exercise history. Then, six volunteers assigned to the control group with notable exercise history were replaced with other volunteers without such exercise history before our main experiment.

#### 3.3.2. Pretest Measurements

Table 1 reports the descriptive statistics related to the demographic and anthropometric characteristics of each of the three groups at the pretreatment stage. In this study, 56 (62%) and four (38%) participants were in their 30s and 40s, respectively. BMI was measured using the on-site InBody or self-reported using an auto-calculator application. BMI is an index that determines whether a person is in the weight range adequate for their height. The normal BMI ranges from 18.5 to 24.9. Most subjects were within the acceptable BMI range.

The participants were equally represented by males and females. The number of males and females was equal for each group. The mean age of males was 37.75 (±5.075), and that of females was 37.95 (±3.362). A total of 60 (67%) participants graduated from a 4 year university, 18 (20%) from a 2 year college, 10 (11%) from high school, and two (2%) from middle school. There were 65 (72%) married participants and 25 (28%) single participants. A total of 19 (21%) people were housewives, 25 (28%) were employed full-time, 14 (16%) worked as freelancers, 19 (21%) were professionals, and 13 (14%) belonged to none of the above.

### 3.4. Main Experiment

The finalized Pilates program was reviewed and confirmed by a Balanced Body master instructor certified by Balanced Body in the US to train instructors and general people. Similarly, a veteran yoga instructor reviewed the finalized yoga program (a director from the local yoga association) with over 10 years of experience. Figure 1 depicts the entire research procedure of this study, the details of which are further described in the subsequent sections.

#### Evaluation of HPLP II and HSRS

All participating individuals were asked to fill in two survey questionnaires, namely, the Health Promoting Lifestyle Profile (HPLP II) and the Health Self-Rating Scale (HSRS) (see Appendix B), upon starting and 1 week after completing their respective exercise program. The same was done for those assigned to the control group (no exercise group).

The reliability of HPLP II in this study, measured with Cronbach’s α, was 0.967. This is well beyond the value of 0.70 suggested for advanced research. Cronbach’s α for each of the six sub-constructs of HPLP II—interpersonal relations, nutrition, health responsibility, physical activity, stress management, and spiritual growth—ranged from 0.827 to 0.914 (see Table 2).

Subjective health status refers to a self-assessment or beliefs about one’s health quality. HSRS consists of three questions using a five-choice Likert scale, with a higher score indicating a more positive rating. The questions in HSRS involve (1) self-assessment according to current physical, physiological, and psychosocial health quality, (2) self-assessment through a historical comparison of health quality relative to 1 year before, and (3) self-assessment through a social comparison of health quality relative to friends and colleagues [37,38].

The reliability of HSRS in this study, measured with Cronbach’s α, was 0.836. This is well beyond the value of 0.70 suggested for advanced research. Cronbach’s α for each of the three subconstructs ranged from 0.714 to 0.897 (see Table 3).

## 4. Statistical Analysis and Results

### 4.1. Statistical Analysis

To test our hypotheses, we analyzed survey responses collected from our participants across the Pilates group, the yoga group, and the control group (no exercise group) using SPSS PASW Statistics 18.00 [39]. Our two dependent variables were (1) post-treatment changes in the participants’ self-reporting of HPLP II and (2) post-treatment changes in the participants’ self-reporting of HSRS. Our hypotheses were deemed as supported if we found statistically meaningful positive changes in post-treatment scores of HPLP II and HSRS among the participants belonging to the Pilates group and the yoga group compared to those in the control group. The Pilates group exhibited the most significant improvements, followed by the yoga group and the control group. 

To see whether these improvements were statistically meaningful, we ran a paired *t*-test in the analysis of the pre- and post-treatment differences with ANOVA [40]. To further check that within- and between-group differences across the three groups are statistically meaningful, we also ran Sheffé test. The Sheffé test, which corrects alpha for multiple mean comparisons, is frequently used to compare mean differences involving more than one pair of means simultaneously [41]. We also employed ANCOVA to further account for the influence of the pretreatment result in our analysis of pre- and post-treatment differences.

### 4.2. Results

#### 4.2.1. Comparison of Pre- and Post-Treatment HPLP II across the Three Groups

The pre- and post-test values were analyzed using one-way ANOVA and a post hoc test, Sheffé test to determine if significant differences in HPLP II values existed among the three groups. Table 4 reports the means and standard deviations of HPLP II for each group at pretreatment and post-treatment. Across the six dimensions and overall HPLP II score, it appears that the Pilates group exhibited the most significant improvements, followed by the yoga group and the control group. Participants who participated in the 8 week yoga program had significantly greater improvements in the nutrition, physical activity, and stress management subscales than seen in the control group, but were not significantly different from the control in the interpersonal relations, health responsibility, or spiritual growth subscales. 

Volunteers who completed the 8 week Pilates program had significantly greater improvements in all six subscales comparted to yoga and control groups. There were statistically significant differences in post-treatment HPLP II among the three groups, as shown in Table 4. Between-group differences in interpersonal relations were statistically significant (F = 12.221, *p* < 0.001). Between-group differences in nutrition were also statistically significant (F = 40.110, *p* < 0.001). Between-group differences in health responsibility were statistically significant as well (F = 19.383, *p* < 0.001). Between-group differences in health responsibility were statistically significant (F = 35.558, *p* < 0.001). Between-group differences in stress management were statistically significant (F = 30.187, *p* < 0.001). Between-group differences in spiritual growth were statistically significant (F = 19.693, *p* < 0.001). Overall, between-group differences in HPLP II at post-treatment were all statistically significant (F = 39.446, *p* < 0.05).

In our pretest analyses of HPLP II, when we checked homogeneity in pretreatment HPLP II across the three groups using one-way ANOVA and Sheffé test, we found statistically significant differences in interpersonal relations and spiritual growth, two sub-constructs of HPLP II. Accordingly, we needed to check further whether these pre-existing differences in Interpersonal relations and spiritual growth values might drive the differences found at post-treatment, using ANCOVA. ANCOVA results are shown in Table 5 and Table 6, which reports significant differences among the three groups for interpersonal relations and spiritual growth, even after accounting for their pretreatment values. The ANCOVA result confirms that these initial differences did not change the relationships. 

#### 4.2.2. Comparison of Pre- and Post-Treatment HSRS across the Three Groups

Table 7 reports the means and standard deviations of HSRS for each group at pretreatment and post-treatment. It seems that the Pilates and yoga group also exhibited a considerable increase in HSRS at post-treatment compared to the control group. Specifically, there was no significant difference across the groups when examining ANOVA results at pretreatment. Notably, there were statistically significant differences in post-treatment HSRS among the three groups, as shown in Table 7. Between-group differences in health status at present were statistically significant (F = 11.140, *p* < 0.001). Between-group differences in health status compared to 1 year prior were also statistically significant (F = 17.522, *p* < 0.001). Between-group differences in health status compared to friends/colleagues (F = 4.915, *p* < 0.001) were statistically significant as well. Overall, between-group differences in HSRS at post-treatment were all statistically significant (F = 3.720, *p* < 0.05).

The pretest score means were evaluated for preexisting variability among the three groups using one-way ANOVA and Sheffé test. This analysis demonstrated that the three groups did not have significant differences in scores for the three subscales at pretest (*p* > 0.05). 

### 4.3. Gender Differences

We further analyzed to see if gender-specific differences existed regarding HPLP II and HSRS scores. Table 8 reports the means and standard deviations of HPLP II for females vs. males at pre- and post-treatment. Gender differences in HPLP II at pretreatment were not statistically meaningful across all subconstructs except for physical activity. Male participants appeared to have reported greater engagement in physical activity, compared to female counterparts, at the pretreatment stage. However, this difference in physical activity did not seem to continue after the 8 week exercise program. Although it is worthwhile to note that males appeared to have reported slightly larger improvements in stress management than females at post-treatment, it is safe to conclude that gender differences in post-treatment changes were not very noticeable in our experimental study as far as the impact of Pilates and yoga on health-promoting behavior is concerned.

Table 9 reports the means and standard deviations of HSRS for females vs. males at pre- and post-treatment, respectively. Gender differences in HSRS at pretreatment were not statistically meaningful across all sub-constructs except for health status at present. Female participants appeared to have reported more positive self-perception about their current health, compared to male counterparts, at pretreatment. However, at post-treatment this difference seemingly disappeared. Again, we did not find much significant gender difference in our experimental study of the impact of Pilates and yoga on other health-promoting behaviors and subjective health status.

## 5. Discussion

The idea that exercise is essential for maintaining a healthy life is almost a truism these days. However, why do some people succeed in adopting a set of health-promoting lifestyle elements, while other people keep failing to stay away from lifestyle elements detrimental to their health? In other words, why do some of them come short of maintaining a healthy lifestyle long enough despite their initial intention to lead a healthy life? Given the importance of this question, understanding the mechanism via which people set in motion the virtual cycle of leading a healthy life is crucial among healthcare professionals and the general public alike. Indeed, it has become increasingly crucial in public healthcare research to have a more nuanced understanding of how different aspects of people’s lives affect their overall wellbeing and the role of physical activity in this process. 

In this study, we examined whether Pilates and yoga participation triggers health-promoting behaviors and positively influences the self-evaluation of health status among the participants. Specifically, we sought to address this question by establishing a causal relationship, not merely a correlation, by comparing the participants’ self-reported changes in health-promoting behaviors—including eating healthy, avoiding a sedentary lifestyle, being responsible for own health, maintaining healthy social relationships, managing stress, and emphasizing spiritual growth—and health status —compared to the prior year and peers—before and after the predesigned 8 week exercise program.

In our analysis of the data collected from 90 female and male adult volunteers, we found that the Pilates group reported the most considerable improvements in the two measures of HPLP II and HSRS, each of which corresponds to self-assessed health-promoting behavior and health status. The increases at post-treatment for the Pilates group appeared even larger than those reported by the yoga group participants. This finding implies that Pilates offers a greater advantage over yoga as an exercise method. The result of this study revealed that engagement in Pilates or yoga for 8 weeks resulted in improved rates of reported engagement in health-promoting behaviors relative to a control group using the HPLP II assessment tool. While improvements in the physical activity subscale were direct and expected, improvements were also seen in the nutrition and stress management subscales for Pilates and yoga and interpersonal relations, health responsibility, and spiritual growth subscales for Pilates to the control. 

Other than stress management, all other HPLP II subscales did not show significant differences in outcomes for males and females. Moreover, HSRS scores showed no significant differences in outcomes between male and female groups. This indicates that the benefits seen from doing Pilates and yoga are not gender-specific and, thus, promotion of these exercise programs should not be gender-restricted. Given that we found no considerable gender differences in the health-promoting impacts of Pilates and yoga, Pilates and yoga can be an effective public health program among both men and women when appropriately implemented.

Despite some similarities, Pilates and yoga do have some notable differences. Here, we briefly discuss some notable differences between Pilates and yoga. First, as depicted in the earlier section, Pilates and yoga have different historical origins, which appear to influence how they have been evolved and organized as a system of fitness program. Pilates is a relatively new invention that started in the early 20th century, while yoga began more than 5000 years ago. Pilates has been successfully evolved as a standardized fitness program and organized as a profession due to its relatively contemporary nature. The formation of the PMA (Pilates Method Association) in 2001, a nonprofit professional association, has accelerated the standardization process by clearly delineating the scope of practice. PMA has also transformed many trainers and instructors into an organized profession by administrating a centralized system of qualification and setting professional standards, as well as establishing a code of ethics in interactions with clients.

In contrast, yoga has variegated and decentralized into numerous branches, which seems to have slowed any standardized and centrally organized attempt. Second, while both Pilates and yoga emphasize harmonization among mind, body, and spirit, their exercise approaches are somewhat different. For example, the breathing techniques utilized differ between Pilates and yoga. Moreover, whereas Pilates tends to focus more on effective muscle energy movement using dynamic, resistance, and stability-related poses, yoga focuses more on how a person feels through relaxation, meditation, and static stretching. Accordingly, Pilates exercises could yield better results more quickly. However, our study has limitations in revealing the mechanisms behind the varied health-related benefits of Pilates and yoga. This topic merits a more detailed investigation in future research with a longer-term intervention.

Our findings are similar to some prior studies, albeit with differences in targeted subjects, which have found associations between engaging in Pilates and yoga exercise programs and perceived health. For instance, in a study by Küçük and Livanelioglu (2015), participating in Pilates led to improvements in healthy women’s perception of exercise and other psychosocial variables [11]. In the study of Berent et al. (2014), yoga was demonstrated to be beneficial for promoting improvements in college students’ quality of life and healthy lifestyle choices [27]. Neumark-Sztainer et al. (2010) examined the associations between participating in mind–body activities such as Pilates and yoga and the prevalence of body dissatisfaction and eating disorders in young adults [28]. In their study, most individuals with body dissatisfaction and eating disorder behaviors had these issues reduced by participation in yoga and Pilates.

However, our findings depart from previous research highlighting Pilates and yoga’s effects on body composition changes and weight loss. In this study, we found only modest decreases in BMI among the Pilates and yoga program participants. This is probably attributed to the fact that our participants’ BMI values were within the average range even before participating in the exercise program. Additional research incorporating participants with obesity would help determine the comparative effectiveness of Pilates vs. yoga on body composition. Pilates focuses more on core control and posture development. In contrast, yoga focuses more on static stretching and flexibility. Therefore, it would also be interesting to see how their effects on body composition might differ among subjects with obesity. Follow-up research can also further examine the health-promoting benefits of Pilates and yoga programs on people with acute or chronic illnesses and any linkage to their rehabilitation.

Given that most people find it challenging to maintain a healthy lifestyle long enough, contrary to their desire to do otherwise, our study provides valuable insights into how to set in motion a virtuous loop of a healthy life. People who practice Pilates or yoga do so not just to increase the amount of their physical activity but also to restore a balance in their hectic everyday lives. This paper has also illuminated one avenue through which Pilates and yoga can help ordinary men and women restore a healthy balance in life. Overall, our findings have useful implications for health policymakers and authorities by offering more clear evidence of Pilates and yoga’s indirect benefits. In this study, Pilates and yoga programs were designed by a master instructor who was certified with an international license and had over 10 years of experience. Those exercises, including goals and precautions, were safe for all people and did not include any inversion exercise or extreme stretching to avoid spine and pelvis issues. Therefore, the implemented programs of Pilates and yoga can be applied to the general public.

This study has a few limitations. The study participants were all ethnic Korean re-siding in South Korea; hence, the results may differ for other ethnicities or locations. This study’s participants were limited to ages 30 to 49, constraining the generalizing of our findings beyond this group. Furthermore, subsequent studies are needed to test how different mixtures of exercise programs in Pilates and yoga might affect health-promoting behavior. Future studies are needed to shed further light on the mechanisms behind the differences between Pilates and yoga. For example, what might be intervening variables in yielding differences in the effects of Pilates and yoga? Given that these two exercises have a slightly different focus, researchers can further examine this question by measuring other factors related to subjects’ self-efficiency and self-control and differences in seeking the balance between physical and mental aspects in life.

## 6. Conclusions

From a public health standpoint, it is essential to ensure that people practice a healthy lifestyle with a belief about its positive effect on their health. Such a lifestyle should include a wholistic approach, taking both physical and psychological aspects of wellbeing seriously [12,28,42,43,44]. In this article, we extend prior research suggesting that various elements of health-promoting behaviors can reinforce one another, highlighting and demonstrating the triggering role of exercise such as Pilates and yoga [12,28,42,43,44]. Overall, our results confirm that Pilates and yoga help recruit health-promoting behaviors in participants and engender positive beliefs about their subjective health status, thereby setting a positive reinforcement cycle in motion. By providing clear evidence that the promotion of Pilates or yoga can serve as an effective intervention strategy that helps individuals change behaviors adverse to their health, this study offers practical implications for health care professionals and public health officials alike. Our work sheds further light on the relationship between participation in low-risk and readily accessible exercise programs such as Pilates and yoga and the likelihood that people engage in comprehensive practices for promoting their health.

## Figures and Tables

**Figure 1 ijerph-18-03802-f001:**
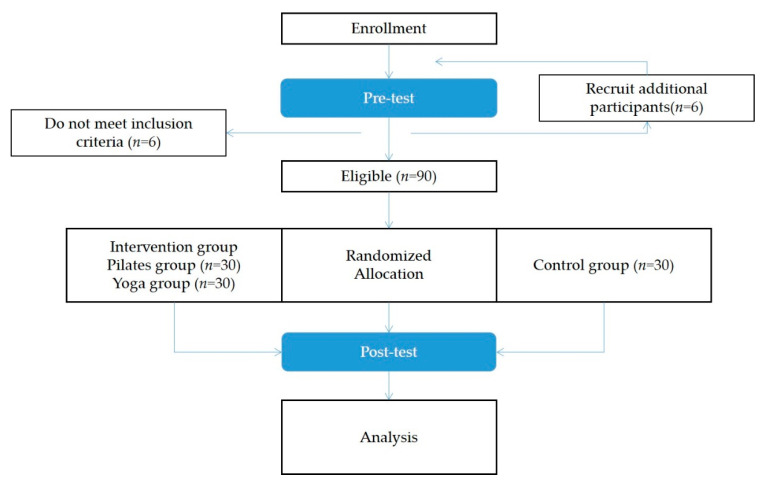
Research procedure.

**Table 1 ijerph-18-03802-t001:** Characteristics of participants at pretreatment.

Characteristics	Control (*n* = 30) Mean ± SD	Pilates (*n* = 30) Mean ± SD	Yoga (*n* = 30) Mean ± SD
Age (years)	35.47 ± 4.562	37.43 ± 4.621	40.67 ± 5.827
Height (cm)	169.97 ± 8.908	167.91 ± 6.127	167.46 ± 8.931
Weight (kg)	64.47 ± 12.942	63.19 ± 10.797	65.07 ± 10.164
Waist circumference (inch)	29.40 ± 3.024	29.93 ± 3.162	29.10 ± 2.496
Hip circumference (inch)	34.53 ± 2.623	34.89 ± 2.347	34.73 ± 2.599
Body mass index (BMI)	22.09 ± 3.050	22.23 ± 3.053	23.09 ± 2.133

**Table 2 ijerph-18-03802-t002:** Reliability of Health Promoting Lifestyle Profile (HPLP) II.

Variables	Survey Item No.	Cronbach’s Alpha
Interpersonal relations	8	0.902
Nutrition	9	0.827
Health responsibility	8	0.900
Physical activity	8	0.914
Stress management	8	0.835
Spiritual growth	9	0.903
HPLP II	50	0.967

**Table 3 ijerph-18-03802-t003:** Reliability of Health Self-Rating Scale (HSRS).

Variables	Cronbach’s Alpha
Health status at present	0.897
Health status compared to one year prior	0.897
Health status compared to friends or colleagues	0.714
HSRS	0.836

**Table 4 ijerph-18-03802-t004:** Pre- and post-treatment HPLP II across the three groups.

Variables	Group	Pre-Test Mean ± SD	Post-Test Mean ± SD	df	F	*p*-Value	Sheffe Test
Interpersonal relations	Control	2.60 ± 0.69	2.46 ± 0.61	88	12.221	0.001 ***	Pilates > yoga, Control
	Pilates	2.47 ± 0.55	3.05 ± 0.41				
	Yoga	2.08 ± 0.46	2.53 ± 0.47				
Nutrition	Control	1.89 ± 0.55	1.78 ± 0.43	88	40.110	0.001 ***	Pilates > yoga > Control
	Pilates	1.90 ± 0.56	2.81 ± 0.48				
	Yoga	1.80 ± 0.37	2.46 ± 0.43				
Health responsibility	Control	1.95 ± 0.80	1.76 ± 0.67	86	19.383	0.001 ***	Pilates > yoga, Control
	Pilates	1.74 ± 0.55	2.64 ± 0.52				
	Yoga	1.68 ± 0.34	2.04 ± 0.44				
Physical activity	Control	1.63 ± 0.62	1.49 ± 0.61	87	35.558	0.001 ***	Pilates > yoga > control
	Pilates	1.70 ± 0.80	2.82 ± 0.61				
	Yoga	1.50 ± 0.40	2.13 ± 0.59				
Stress management	Control	1.95 ± 0.54	1.83 ± 0.44	89	30.187	0.001 ***	Pilates > yoga > control
	Pilates	1.90 ± 0.58	2.73 ± 0.48				
	Yoga	1.75 ± 0.39	2.23 ± 0.42				
Spiritual growth	Control	2.45 ± 0.67	2.28 ± 0.58	88	19.693	0.001 ***	Pilates > yoga, control
	Pilates	2.37 ± 0.55	2.99 ± 0.33				
	Yoga	1.98 ± 0.37	2.40 ± 0.46				
HPLP II	Control	2.04 ± 0.50	1.90 ± 0.42	81	39.446	0.001 ***	Pilates > yoga > control
	Pilates	2.02 ± 0.5	2.85 ± 0.40				
	Yoga	1.79 ± 0.33	2.30 ± 0.38				

*** *p* < 0.001.

**Table 5 ijerph-18-03802-t005:** The ANCOVA result of interpersonal relations.

Classification	SS	df	MS	F	*p*-Value
Pretreatment	10.187	1	10.187	75.825	0.001 ***
Interpersonal relations	6.357	2	3.178	23.657	0.001 ***
Error	11.420	85	0.134		
Total	27.749	88			

*** *p* < 0.001, Note: SS stands for sum of squares; MS stands for mean square.

**Table 6 ijerph-18-03802-t006:** The ANCOVA result of spiritual growth.

Classification	SS	df	MS	F	*p*-Value
Pretreatment	7.327	1	7.327	53.389	0.001 ***
Spiritual Growth	8.489	2	4.244	30.927	0.001 ***
Error	11.665	85	0.137		
Total	27.690	88			

*** *p* < 0.001, Note: SS stands for sum of squares; MS stands for mean square.

**Table 7 ijerph-18-03802-t007:** Pre- and post-treatment HSRS across the three groups.

Items	Group	Pre-Test Mean ± SD	Post-Test Mean ± SD	df	F	*p*-Value	Sheffe Test
Health status at present	Control	2.77 ± 0.935	2.67 ± 0.884	89	11.140	0.001 ***	Pilates, yoga > control
	Pilates	2.70 ± 0.988	3.43 ± 0.971				
	Yoga	3.03 ± 0.490	3.63 ± 0.615				
Health status compared to one year prior	Control	2.63 ± 0.615	2.57 ± 0.626	89	17.522	0.001 ***	Pilates > yoga > control
	Pilates	2.62 ± 0.775	3.79 ± 0.978				
	Yoga	3.00 ± 0.263	3.13 ± 0.571				
Health status compared to friends/colleagues	Control	3.04 ± 0.815	2.97 ± 0.687	89	4.915	0.001 ***	Pilates, yoga > control
	Pilates	3.08 ± 0.725	3.80 ± 0.610				
	Yoga	3.11 ± 0.320	3.53 ± 0.514				
HSRS	Control	2.81 ± 0.786	2.73 ± 0.732	88	3.720	0.028 *	Pilates, yoga > control
	Pilates	2.92 ± 0.829	3.67 ± 0.853				
	Yoga	3.04 ± 0.357	3.43 ± 0.566				

*** *p* < 0.001, * *p* < 0.05.

**Table 8 ijerph-18-03802-t008:** Comparison of male and female groups for HPLP II.

Variables	Male		Female		T		*p*-Value	
	Pretest Mean ± SD	Post-Test Mean ± SD	Pretest Mean ± SD	Post-Test Mean ± SD	Pretest	Post-Test	Pretest	Post-Test
Interpersonal relations	2.49 ± 0.609	2.72 ± 0.513	2.28 ± 0.592	2.64 ± 0.608	1.712	0.740	0.565	0.461
Nutrition	1.90 ± 0.535	2.33 ± 0.604	1.83 ± 0.461	2.38 ± 0.633	0.632	−0.377	0.710	0.707
Health responsibility	1.89 ± 0.637	2.26 ± 0.675	1.69 ± 0.553	2.04 ± 0.640	1.612	1.536	0.384	0.128
Physical activity	1.83 ± 0.765	2.25 ± 0.745	1.40 ± 0.348	2.03 ± 0.864	3.468	1.285	0.000 ***	0.202
Stress management	2.03 ± 0.548	2.38 ± 0.551	1.70 ± 0.412	2.14 ± 0.586	3.235	2.039	0.105	0.044 *
Spiritual growth	2.39 ± 0.587	2.56 ± 0.512	2.14 ± 0.548	2.56 ± 0.613	2.019	−0.063	0.376	0.950
HPLP II	2.07 ± 0.506	2.41 ± 0.531	1.84 ± 0.417	2.31 ± 0.599	2.270	0.735	0.175	0.465

*** *p* < 0.001, * *p* < 0.05.

**Table 9 ijerph-18-03802-t009:** Comparison of male and female groups for HSRS.

Variables	Male		Female		T		*p*-Value	
	Pretest Mean ± SD	Post-Test Mean ± SD	Pretest Mean ± SD	Post-Test Mean ± SD	Pretest	Post-Test	Pretest	Post-Test
Health status at present	3.11 ± 0.982	3.18 ± 0.936	3.22 ± 0.670	3.31 ± 0.925	−0.627	−0.680	0.021 *	0.499
Health status compared to one year prior	3.18 ± 0.650	3.16 ± 0.878	3.32 ± 0.561	3.13 ± 0.919	0.758	0.117	0.095	0.907
Health status compared to friends/colleagues	2.84 ± 0.903	3.44 ± 0.661	2.96 ± 0.903	3.43 ± 0.780	3.810	0.027	0.054.	0.978
HSRS	3.04 ± 0.845	3.26 ± 0.825	3.16 ± 0.711	3.29 ± 0.356	0.830	−1.011	0.046	0.830

* *p* < 0.05.

## Data Availability

Data supporting reported results are available from the authors upon request.

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
