# Peer review of "The Impacts of Pilates and Yoga on Health-Promoting Behaviors and Subjective Health Status"

_ijerph, 2021, doi:10.3390/ijerph18073802_

Round 1

Reviewer 1 Report

GENERAL CONSIDERATION

The article is interesting, even though it is not clear if previous studies already assess the same topics and outcomes.

Moreover, lots of improvements should be considered to improve the manuscript's quality. The manuscript is repetitive in some parts, reporting unnecessary details in others, missing some important details about the sample and the procedure, and the required chapter organization. In particular, the disorganized methods section makes it difficult to read and comprehend the results section.

Finally, some clarifications or additional statistical analysis could be considered.

Please, see below for detailed comments.

TITLE

What does "other" in the title mean?

ABSTRACT

  1. Line 11: "Still,…" this sentence is superfluous because, in the paper, the authors did not analyze the causality.
  2. Some details are missing, mean age, sex composition, utilized tests.

INTRODUCTION

  1. From lines 83 to lines 98, the authors describe the aims and some details of the sample, but then they turn again to speak about the introduction with Pilates and yoga characteristics. In line 218, the study's aim is explained differently from before, and two hypotheses are introduced. In my opinion, it should be better to completely cover the introduction section at first and, at the end of the introduction, explain and motivate the study's aim. Participants, statistical analysis, and procedures should all be inserted in the Material and methods section with appropriate heading and subheadings.
  2. I suggest two articles on physical activity, yoga, and sleep that can be further considered in the introduction section: doi:10.3390/nu12123622 and doi:10.1200/JCO.2012.43.7707.
  3. The introduction is very interesting and exhaustive; however, I am wondering if it is too long. The audience is supposed to already know much of the provided information.
  4. In my opinion, chapter 2 should be merged with chapter 1 (introduction), and 2.1 and 2.2 are two subheadings of the introduction.
  5. Lines 199-217: these topics are not related to yoga or pilates; they are general considerations.

METHODS

  1. In my opinion, this section should be re-organized entirely by considering the introduction of participants, study design, and statistical sections.
  2. The authors should be more synthetic and more focused to the point: they repeat a lot of time the protocol, the length of the study, etc. By organizing better the methods section, such repetitions won't be needed.
  3. Lots of important information are missing, such as inclusion and exclusion criteria, place of recruitment, percentage sex composition, percentage sex composition of each group, number of ethical committee approbation, etc.
  4. Lines 234-240: study aims for the third time.
  5. Please explain the second questionnaire while the first is fully described; mixing the two descriptions is very confusing.
  6. Pilot test: without a detailed explanation of the study protocol, it isn't easy to understand the study's structure and phases.
  7. Main experiment: please be more concise and avoid useless details.
  8. Line 299: these are anthropometric characteristics and not physiological.
  9. Line 366, Fortunately is not the appropriate word.
  • It is not clear how the authors verified previous or actual physical activity practice and if they include only physical activity naïve.
  • Lines 383-404: these are results
  1. α value is missing, normal distribution assessment is missing, eventual covariates are missing. Please write an appropriate statistical analysis section.

RESULTS

  1. Without a proper statistical analysis section, it isn't effortless to follow the results.
  2. Lines 414-416: the software should be named in the statistical section.
  3. Are all the tables necessary? Might the authors consider showing some of them as supplementary material?
  4. Table 4: did the authors test differences?
  5. Line 432: what is on-site Inbody?
  6. Line 444-445: did the authors run a t-test and ANOVA at the same time? Which test did they use?
  7. Might the authors try to merge tables 5 and 6? Not all ANOVA indexes should be reported (usually only df, F, and p), and in this way, mean values and significances could be together and easily to understand. Obviously, this should be applied to all other tables.
  8. The authors used to say "significant differences" without explaining them, i.e., who collected the best values? Who improved most? What do these differences mean?
  9. Does M represent the mean values? It should be clarified.
  10. The tables' legends should be improved; sometimes they are missing, and other times they explained symbols not reported in the table.
  11. Did the authors adjust the analysis for sex and age?
  12. In the gender analysis, did the author take into consideration the physical activity program or group?

DISCUSSION

  1. The discussion should be focused on the results and not on the origins of Pilates and Yoga. The authors should consider revising it in light of the results and try to explain them, considering previous results.
  2. In the title, the authors asked a question. Does the discussion answer the question?
  3. The limitation section is missing.
  4. The conclusions should be more concise, just one paragraph with the main results, future perspectives, or practical applications.

Author Response

Response to Reviewer 1

We have revised the manuscript in response to the helpful and detailed comments you provided. In our view, this revision has led to a great deal of improvement, further strengthening our paper. We are very grateful for your feedback that has led to these improvements. In the detailed response below, we describe our efforts to respond to the concerns and questions raised. We hope you will find the new manuscript clearer and of better quality.

Below are our point-by-point responses to each of your comments.

Title

What does "other" in the title mean?

Given that Pilates and yoga are also a subset of health-promoting behaviors, we initially thought that by adding ‘other’ we could emphasize a kind of spillover influence of Pilates and Yoga on ‘other’ health-promoting behaviors.

But on the second thought, we now think that adding “other” could cause unnecessary confusion for readers. So the title is corrected as follows. And 'other' was deleted from the main part as well.

We appreciate your clarifying question.

The Impacts of Pilates and Yoga on Health-Promoting Behaviors and Subjective Health Status

Abstract

1. Line 11: "Still,…" this sentence is superfluous because, in the paper, the authors did not analyze the causality.

Thank you for pointing this out. We agree that our study does not sufficiently establish the causality between Pilates and yoga and health-promoting behaviors and Subjective Health Status. Therefore, this sentence was eliminated. We also revised our entire manuscript to reflect this change.

2. Some details are missing, mean age, sex composition, utilized tests

Based on your suggestion, we have revised our Abstract and included the information about the subjects’ age, sex composition, and tests deployed (yellow highlighted) in this Abstract as below.

Abstract: This study investigates whether Pilates and yoga lead people to adopt generally health-promoting lifestyle elements and feel better about their physical and mental fitness. To this end, we designed an 8-week exercise program of Pilates and yoga reviewed by veteran practitioners and conducted an experimental study through which we collected the data from 90 volunteered adult subjects between ages 30 and 49 (mean age=35.47), equally represented by women and men without previous experience with Pilates or yoga. In the 8-week long experiment, we assigned the subjects to three groups, where subjects in the two exercise groups regularly took part in either Pilates or yoga classes, and the control group participated in neither exercise classes. All participants completed two surveys, the Health-Promoting Lifestyle Profile (HPLP II) and the Health Self-Rating Scale (HSRS), before and after their assigned program. In our analysis of pre- and post-treatment differences across the three groups, we ran ANOVA, ANCOVA, and Sheffe’s test implemented using SPSS PASW Statistics 18.00. Our results indicate that Pilates and yoga groups exhibited a higher engagement in health-promoting behaviors than the control group after the program. Subjective health status, measured with HSRS, also improved significantly among Pilates and yoga participants compared to those in the control group after the program. The supplementary analysis finds no significant gender-based difference in these impacts. Overall, ….

Introduction

1. From lines 83 to lines 98, the authors describe the aims and some details of the sample, but then they turn again to speak about the introduction with Pilates and yoga characteristics. In line 218, the study's aim is explained differently from before, and two hypotheses are introduced. In my opinion, it should be better to completely cover the introduction section at first and, at the end of the introduction, explain and motivate the study's aim. Participants, statistical analysis, and procedures should all be inserted in the Material and methods section with appropriate heading and subheadings.

Thank you for these comments and suggestions. We have rewritten and reorganized this part (see below).

2. I suggest two articles on physical activity, yoga, and sleep that can be further considered in the introduction section: doi:10.3390/nu12123622 and doi:10.1200/JCO.2012.43.7707.

Thank you for introducing additional related work. We have reviewed those articles and cited them in our Introduction (see below, highlighted in yellow).

3. The introduction is very interesting and exhaustive; however, I am wondering if it is too long. The audience is supposed to already know much of the provided information.

We have shortened this part, removing the unnecessary details (see below).

4. In my opinion, chapter 2 should be merged with chapter 1 (introduction), and 2.1 and 2.2 are two subheadings of the introductions

Thank you for this idea. However, we thought that as you pointed out the above, the Introduction is too long as it stands. Instead, we decided that chapter 2 is divided into two subsections as below.

  1. Research Background and Hypotheses

2.1. The Origins and Principles of Pilates and Yoga

2.2. The Impacts of Pilates and Yoga on Health-Promoting Behavior and Subjective Health Status

5. Lines 199-217: these topics are not related to yoga or pilates; they are general considerations.

We agree that covering prior studies of little direct relevance to Pilates and yoga could distract audience attention. Therefore, we decided to remove these paragraphs. We appreciate your suggestions on strengthening our Introduction. Following your advice from 1~3 and 5, we revised our Introduction as follows:

  1. Introduction

Less known, however, is whether Pilates and yoga may also have secondary, indirect benefits. Adopting a broad set of lifestyle elements that are generally regarded as beneficial to one’s physical and psychological health and holding a positive belief about one’s health might be necessary for leading a healthy life. Some prior research hints at the possibility that one form of health-promoting behavior triggers people to adopt broadly health-promoting lifestyle elements (e.g., healthy eating, enriching interpersonal relationships, informed health-related decisions, work-life balance, emphasis on spiritual growth, etc.), thereby setting in motion a positive reinforcement loop. For example, Kucuk and Livanelioglu (2015) found that individuals starting a clinical Pilates program tend to hold a positive notion about the goodness of exercises and experience an improvement in their socio-psychological aspects of lives [11]. Mustian’s research team (2013) found that yoga has a positive impact on the sleep quality among cancer survivors [13]. Galasso and colleagues (2020) found that aerobic and anaerobic exercise training was effective in reducing the tendency of binge eating [14].

However, these studies come short of providing more comprehensive and clear evidence that Pilates leads to a positive change in both the participants’ overall health-promoting behavior and self-perceived health status. Besides, Kucuk and Livanelioglu (2015)’s study has primarily targeted people taking a clinical Pilates program, excluding those who might regularly engage in Pilates for non-clinical purposes [11]. And Pilates was not covered in studies by Mustian et al. (2013) and Galasso et al. (2020) [13, 14]. Given this, the current study seeks to fill this deficiency in prior work by directly investigating whether regularly engaging in an exercise program such as Pilates and yoga might positively affect participants’ adoption of health-promoting behaviors. Additionally, we examine the extent to which these outcomes shown in program participants are accompanied by an increase in self-perceived physical and mental well-being.

Methods

1. In my opinion, this section should be re-organized entirely by considering the introduction of participants, study design, and statistical sections. The authors should be more synthetic and more focused to the point: they repeat a lot of time the protocol, the length of the study, etc. By organizing better the methods section, such repetitions won't be needed. Lots of important information are missing, such as inclusion and exclusion criteria, place of recruitment, percentage sex composition, percentage sex composition of each group, number of ethical committee approbation, etc.

Thank you for your suggestions. Following your advice, we did completely reorganize our method section. We also included the information you pointed out as missing while revising. (see below, highlighted in yellow)

2. Please explain the second questionnaire while the first is fully described; mixing the two descriptions is very confusing.

Thank you for your suggestions. We used a more clear structure in describing these two measures. Also we provided more details for the second questionnaire (see below, highlighted in yellow).

3. Pilot test: without a detailed explanation of the study protocol, it isn't easy to understand the study's structure and phases.

We provided a more clear explanation for the pilot test (see below, highlighted in yellow).

4. Main experiment: please be more concise and avoid useless details.

Following your advice, we revised this section trying to be more concise in our description. (see below, highlighted in yellow).

5. Line 299: these are anthropometric characteristics and not physiological.

We corrected this. Thank you.

6. Line 366, Fortunately is not the appropriate word.

We corrected this. Thank you.

7. It is not clear how the authors verified previous or actual physical activity practice and if they include only physical activity naïve.

Thank you for asking. The authors and instructors did their best to regularly check in with subjects in the control group to verify their exercise status.

8. Lines 383-404: these are results

We corrected this. Thank you.

9. α value is missing, normal distribution assessment is missing, eventual covariates are missing. Please write an appropriate statistical analysis section.

Following your advice, we provided revised tables of our statistical analysis. P-values are provided throughout these tables.

  1. Research Method

To test our hypotheses, we designed an experiment where our treatment is Pilates and yoga program intervention, and the two dependent variables of our interest are (1) post-treatment changes in the participants’ self-reporting of engagement in health-promoting behaviors (HPLP II) and (2) post-treatment changes in the participants’ self-reporting of subjective health status (HSRS). Our analytical strategy compares pre-treatment and post-treatment changes across the three groups, including the Pilates group, the yoga group, and the control group (no exercise group).

In October 2017, the first author obtained approval for this experiment in which human subjects were involved from the Institutional Review Board (IRB) of the Korean National Sports University, with which the first author was then affiliated.

3.1. HPLP II and HSRS

The Health Promoting Lifestyle Profile (HPLP) is a widely used assessment tool first developed by Walker, Sechrist, & Pender (1987, 1995) [15,16]. In their early work, these researchers focused primarily on developing psychometric scales [15]. Later, the original HPLP was updated to reflect a comprehensive nature of a person’s behaviors that has ramifications for her physical and mental fitness and evolved as the HPLP II [31].

HPLP II uses the four-point Likert scale with higher scores indicating greater participation in health-promoting activities and consists of 52 items that can be divided into six component clusters—(1) health responsibility, (2) physical activity, (3) nutrition, (4) spiritual growth (or self-actualization), (5) interpersonal relations (or interpersonal support), and (6) stress management [31]. Health responsibility captures how a person pays attention to and takes good care of her health based on health professionals’ guides. Physical activity is a measure of whether a person does work-out of varying intensity and engages in physical activity of varying kinds and levels. Nutrition captures the extent to which a person consumes appropriate quality and quality foods. Spiritual growth measures the degree to which a person nurtures inner and leads a meaningful life. Interpersonal relations reflect how a person cares about maintaining a meaningful relationship with others and can connect socially and emotionally. Stress management measures the extent to which a person can release her stress and keep stressors in control [16]. The HPLP II has been deployed extensively by researchers interested in the relationships among these sub-scales and the impacts of a specific physical activity [17-22].

The Health Self-Rating Scale (HSRS) is a comprehensive self-assessment of health quality. HSRS was first developed by Lawton et al. (1982) and has been widely adopted across numerous studies on subjective health status. HSRS survey asks participants to rate their present health status compared to how it was one year prior and how it compares to their friends or colleagues[25, 26].

This study adopted a local adaptation of the original HPLP II and the original HSRS. We relied on Korean translations of HPLP II and HSRS performed by Seo and Hah (2004) and Shin and Kim (2009), respectively [25, 26].

3.2. Intervention Program

Two Pilates and two yoga experts …

3.2.1. Pilot Test

In November 2017, a pilot test was started with nine people who had participated in a 4-week exercise program. Our 4-week pilot study aimed to ensure the appropriate design for our main experiment to be followed. Our focus was to configure the exercise program's adequate task complexity levels in advance of the full-scale experiment to last over eight weeks.

The pilot test subjects were informed that they were participating in some study on a community health program and that their personal information would be kept confidential. Participants were randomly assigned to each of the three groups: a 4-week Pilates program, a 4-week yoga program, or the control group with no specified exercise program.

During the pilot study, we noticed that some of the exercises we initially included in the program were quite challenging and problematic for some people. For example, the roll-over movement in Pilates turned out to be rather hard for those with no prior exposure to Pilates and yoga. It was crucial to ensure that the exercises included in each program were safe and secure, and suitable for beginning and intermediate-level students who do not have any Pilates or yoga experience. Given that people usually have a weak core and unstable lumbopelvic stability, the rolling-over, inversion movements, or extreme stretching were removed to minimize any injury or accident during the sessions.

3.3. Participants

Our inclusion criteria were age 30-49 years, no severe disease, and have interest in yoga and Pilates. Our exclusion criteria were that having prior experience for yoga and Pilates and currently participating regularly in exercises such as swimming and regular fitness work-outs.

Yoga Community and Korea Pilates …

3.3.1. Pre-screening

A total of 90 volunteers participated in this project. All participants took their first Pilates or yoga class or did not have prior experience with Pilates or yoga. We provided them with a privacy statement stipulating that their personal information would remain strictly confidential.

3.3. Main Experiment

The finalized Pilates program was reviewed and confirmed by a Balanced Body master instructor certified by Balanced Body in the USA to train instructors and general people. Similarly, a veteran yoga instructor reviewed the finalized yoga program (a director from the local yoga association) with over ten years of experience. Figure 1 depicts the entire research procedure of this study, of which details are further described in the subsequent sections.

Results

Please refer to the resubmitted manuscript for extensive revision.

1. Lines 414-416: the software should be named in the statistical section.

We corrected this. Thank you.

3. Are all the tables necessary? Might the authors consider showing some of them as supplementary material?

Following your advice, we provided these tables as supplements. 

4. Table 4: did the authors test differences?

Thank you for your question. We provide tests on this in Table 4.

5. Line 432: what is on-site Inbody?

Thank you for asking. InBody is a branded BMI equipment.

6. Line 444-445: did the authors run a t-test and ANOVA at the same time? Which test did they use?

Thank you for your question. It is explained in the 4.1. Statistical Analysis section.

7. Might the authors try to merge tables 5 and 6? Not all ANOVA indexes should be reported (usually only df, F, and p), and in this way, mean values and significances could be together and easily to understand. Obviously, this should be applied to all other tables.

Thank you for your comments. We have reorganized and replaced the all tables.

8. The authors used to say "significant differences" without explaining them, i.e., who collected the best values? Who improved most? What do these differences mean?

Thank you for your comments. We have provided detailed explanations in the Results.

9. Does M represent the mean values? It should be clarified.

Yes. We have used ‘Means’ in all of the Tables.

10. The tables' legends should be improved; sometimes they are missing, and other times they explained symbols not reported in the table.

Thank you for your comments. All tables were revised.

11. Did the authors adjust the analysis for sex and age?

Thank you for your question. Our participants were selected from 30 to 49 to narrow the difference. Gender ratio in our subjects was 50:50. Accordingly, in this random assignment experiment, there is little need for adjustment for these factors.

12. In the gender analysis, did the author take into consideration the physical activity program or group?

Thank you for your question. Like HPLP II and HSRS assessment, gender comparison was performed through the experiment with Pilates and yoga.

Discussion

1. The discussion should be focused on the results and not on the origins of Pilates and Yoga. The authors should consider revising it in light of the results and try to explain them, considering previous results.

Thank you for your comments. Following your advice, we have extensively revised our Discussion.

2. In the title, the authors asked a question. Does the discussion answer the question?

We believe so in this revised version.

3. The limitation section is missing.

We stated several limitations of our study. (see below, highlighted in yellow).

  1. Discussion

Other than Stress Management, all other HPLP II subscales did not show significant differences in outcomes for males and females. Also, HSRS scores showed no significant differences in outcomes between male and female groups. This indicates that the benefits seen from doing Pilates and yoga are not gender-specific, and thus promotion of these exercise programs should not be gender restricted. Given that we found no considerable gender differences in the health- promoting impacts of Pilates and yoga, Pilates and yoga can be an effective public health program among both men and women when appropriately implemented.

This study has a few limitations. The study participants were all ethnic Korean residing in South Korea, so that the results may differ for other ethnicities or locations. This study's participants were limited to ages 30 to 49, constraining the generalizing our findings beyond this group. Also, subsequent studies are needed to test how different mixtures of exercise programs in Pilates and yoga might affect health-promoting behavior.

4. The conclusions should be more concise, just one paragraph with the main results, future perspectives, or practical applications.

Thank you for your comments. Following your advice, we have revised this section trying to be more concise in our description. (see below.)

  1. Conclusions

From a public health standpoint, it is essential to ensure that people practice a healthy lifestyle with a belief about its positive effect on their health. Such a lifestyle should include a wholistic approach, taking both physical and psychological aspects of well-being seriously [12, 46-49]. In this article, we have extended prior research suggesting that various elements of health-promoting behaviors can reinforce one another, high-lighting and demonstrating the triggering role of exercise such as Pilates and yoga [12, 46-49]. Overall, our results confirm that Pilates and yoga help recruit health-promoting behaviors in participants and engender positive beliefs about their subjective health status, thereby setting a positive reinforcement cycle in motion. By providing clear evidence that the promotion of Pilates or yoga can serve as an effective intervention strategy that helps individuals change behaviors adverse to their health, this study offers practical implications for health care professionals and public health officials alike. Our work has shed further light on the relationship between participation in such low-risk and readily accessible exercise programs as Pilates and yoga and the likelihood that people engage in comprehensive practices of promoting their health.

Reviewer 2 Report

Thank you for carrying out this research work

Indicate that its format does not correspond to what is needed to be an article, it is more similar to a final degree project.

The quotes used must be more current line 43 quote 3 of (1895)

Line 43- There are statements not based on evidence. "most doctors ...."

Line 46, 48, 64, 71 which is indicated based on very old evidence

Line 94.95, the scale references must indicate data on the validity and reliability of said scales.

From 99 to 100, where it is indicated "many people want to lead a healthy life ...". I do not know where this data is obtained, has a scale been passed to determine that there are many or few? what value is a lot, or a little? Has any stopover been passed? What is a healthy life?

from 108 to 144 an unnecessary summary is made

From line 100 to 105- everything that is indicated does not apply for an article

Line 166 - backed by outdated citations

line 185- not very technical language and writing

line 217 to 244- all repeated. It is not necessary

line 257- the important thing is not that it is used a lot, the important thing is its characteristics.

Line 301- specify how informed consent is, if it meets all ethical requirements.

line 352- a schedule had been necessary

The tables with the description of the sessions, are in annexes

The results do not need to give such explanations. Doesn't follow a format for an article

Author Response

Response to Reviewer 2

We have revised the manuscript in response to the helpful and detailed comments you provided. In our view, this revision has led to a great deal of improvement, further strengthening our paper. We are very grateful for your feedback that has led to these improvements. In the detailed response below, we describe our efforts to respond to the concerns and questions raised. We hope you will find the new manuscript clearer and of better quality.

Below are our point-by-point responses to each of your comments.

Introduction

1. The quotes used must be more current line 43 quote 3 of (1895)

Thank you for your suggestions. Following your advice, we cited current citations in our Introduction (see below, highlighted in yellow).

2. Line 43- There are statements not based on evidence. "most doctors ...."

Thank you for these comments and suggestions. We have rewritten and reorganized Introduction part (see below).

3. Line 46, 48, 64, 71 which is indicated based on very old evidence

Thank you for these comments and suggestions. We have rewritten and reorganized Introduction part (see below).

4. From 99 to 100, where it is indicated "many people want to lead a healthy life ...". I do not know where this data is obtained, has a scale been passed to determine that there are many or few? what value is a lot, or a little? Has any stopover been passed? What is a healthy life?

Thank you for these comments. We have rewritten Introduction part (see below).

5. From 108 to 144 an unnecessary summary is made

Thank you for your comments. We have rewritten Introduction part (see below)..

6. From line 100 to 105- everything that is indicated does not apply for an article

Thank you for your comments. We have rewritten Introduction part (see below).

7. Line 166 - backed by outdated citations

Thank you for your comments. Following your advice, we cited current citations in our Introduction (see below, highlighted in yellow).

8. line 185- not very technical language and writing

Thank you for your comments. We have rewritten Introduction part (see below).

  1. Introduction

Exercise is well documented as having various health benefits. Abundant evidence suggests that regular physical activity is associated with lower obesity rate and cardiovascular disease incidence, better sleep patterns and sexual function, and slower aging-related deterioration of the immune system [1,2]. Individuals who exercise routinely also tend to report better mood states than those who do not [1]. Indeed, exercise is generally accepted as an integral component of health-promoting behavior, defined as a broad set of lifestyle elements—including consuming nutritious foods, maintaining adequate sleep, minimizing stress, and staying away from health-detrimental habits such as smoking—that is positively relate to better health [3]. Most physicians subscribe to the idea that exercise helps mitigate the risk of disease, prevent premature death, and improve the overall quality of life. From a public health standpoint, exercise is one of the most cost-effective means by which public health goals can be attained [3]. The notion that exercise is essential for leading a healthy life, both physically and psychologically, is now indisputable.

Among many kinds of physical activity programs, it is noteworthy that Pilates and yoga have gained increasing popularity amongst the general public over the past two decades. Pilates and yoga are particularly appealing due to their direct benefits on physical well-being—including weight control and improved posture, flexibility, and cardiovascular function—that come with low risks of sports-related injuries [3]. According to an annual survey conducted by IDEA Fitness Programs and Equipment Survey in 2007, Pilates ranked sixth on the most frequently offered exercise programs, a vast improvement since 1999 [4]. In the same year, Yoga also ranked 13th though its position had undergone gradual declines from its peak in 2002. In annual Fitness Trends Surveys carried out by a US-based association of Sports Medicine, Pilates and yoga have been frequently listed as the Worldwide Fitness Trends since 2008 [5].

Evidence of Pilates and yoga's direct health benefits is growing. For example, some studies show that regular engagement in Pilates is associated with a boost in functional autonomy, balance, flexibility, and muscle strength [6-12]. Other studies show that regular yoga participation helped individuals alleviate muscle-related pains, especially among adults with sedentary lifestyles or suffering from chronic illnesses [9, 10].

Less known, however, is whether Pilates and yoga may also have secondary, indirect benefits. Adopting a broad set of lifestyle elements that are generally regarded as beneficial to one’s physical and psychological health and holding a positive belief about one’s health might be necessary for leading a healthy life. Some prior research hints at the possibility that one form of health-promoting behavior triggers people to adopt broadly health-promoting lifestyle elements (e.g., healthy eating, enriching interpersonal relation-ships, informed health-related decisions, work-life balance, emphasis on spiritual growth, etc.), thereby setting in motion a positive reinforcement loop. For example, Kucuk and Livanelioglu (2015) found that individuals starting a clinical Pilates program tend to hold a positive notion about the goodness of exercises and experience an improvement in their socio-psychological aspects of lives [11]. Mustian’s research team (2013) found that yoga has a positive impact on the sleep quality among cancer survivors [13]. Galasso and col-leagues (2020) found that aerobic and anaerobic exercise training was effective in reducing the tendency of binge eating [14].

However, these studies come short of providing more comprehensive and clear evidence that Pilates leads to a positive change in both the participants’ overall health-promoting behavior and self-perceived health status. Besides, Kucuk and Livanelioglu (2015)’s study has primarily targeted people taking a clinical Pilates pro-gram, excluding those who might regularly engage in Pilates for non-clinical purposes [11]. And Pilates was not covered in studies by Mustian et al. (2013) and Galasso et al. (2020). Given this, the current study seeks to fill this deficiency in prior work by directly investigating whether regularly engaging in an exercise program such as Pilates and yoga might positively affect participants’ adoption of health-promoting behaviors. Additionally, we examine the extent to which these outcomes shown in program participants are ac-companied by an increase in self-perceived physical and mental well-being.

Methods  

1. line 217 to 244- all repeated. It is not necessary

Following your advice, we revised this section trying to be more concise in our description. (see below).

2. line 257- the important thing is not that it is used a lot, the important thing is its characteristics.

Following your advice, we revised this section trying to be more concise in our description. (see below).

3. Line 94.95, the scale references must indicate data on the validity and reliability of said scales.

Following your advice, we included the information you pointed out (see below, highlighted in yellow).

4. Line 301- specify how informed consent is, if it meets all ethical requirements.

Thank you for your comments. We provided an explanation (see below, highlighted in yellow).

5. line 352- a schedule had been necessary

Thank you for your comments. We put detailed schedule in Method. For your information, here is a summary of our schedule:

Pilot test : Novemver 1st to 30th , 2017

Training instructors : December 1st to 6th, 2017

Orientation & Pre-test : December 7th, 2017

Main experiment (exercise programs): December 12th, 2017 to February 1st, 2018

Post-test : February 8th, 2018

6. The tables with the description of the sessions, are in annexes

Thank you for your suggestions. Following your advice, we provided these tables in Appendix A.

  1. Research Method

To test our hypotheses, we designed an experiment where our treatment is Pilates and yoga program intervention, and the two dependent variables of our interest are (1) post-treatment changes in the participants’ self-reporting of engagement in health-promoting behaviors (HPLP II) and (2) post-treatment changes in the participants’ self-reporting of subjective health status (HSRS). Our analytical strategy compares pre-treatment and post-treatment changes across the three groups, including the Pilates group, the yoga group, and the control group (no exercise group).

In October 2017, the first author obtained approval for this experiment in which hu-man subjects were involved from the Institutional Review Board (IRB) of the Korean National Sports University, with which the first author was then affiliated.

3.3.1. Pre-screening

A total of 90 volunteers participated in this project. All participants took their first Pilates or yoga class or did not have prior experience with Pilates or yoga. We provided them with a privacy statement stipulating that their personal information would remain strictly confidential.

3.3.1. Evaluation of HPLP II and HSRS

All participating individuals were asked to fill in the survey questionnaire twice, including the Health Promoting Lifestyle Profile (HPLP II) and the Health Self-Rating Scale (HSRS) (see Appendix B), starting and one week after completing their respective exercise program. The same was done for those assigned to the control group (no exercise group).

The reliability of HPLP II in this study, measured with Cronbach's α, was 0.967. This is well beyond 0.70 suggested for advanced research. Cronbach's α for each of the six sub-constructs of HPLP II—Interpersonal Relations, Nutrition, Health Responsibility, Physical Activity, Stress Management, and Spiritual Growth— ranged from 0.827 to 0.914 (see Table 2).

Subjective health status refers to a self-assessment or beliefs about one’s health quality. HSRS consists of three questions using the five-choice Likert scale, with a higher score indicating more positive rating. Three questions in HSRS are (1) self-assessment about current physical, physiological, and psychosocial health quality; (2) self-assessment about historical comparison of health quality relative to a year before; and (3) self-assessment about social comparison of health quality relative to friends and colleagues [25, 26].

The reliability of HSRS in this study, measured with Cronbach's α, was 0.836. This is well beyond 0.70 suggested for advanced research. Cronbach's α for each of the three sub-constructs ranged from 0.714 to 0.897 (see Table 3).

Results

1. The results do not need to give such explanations. Doesn't follow a format for an article

Thank you for your comments. We did reorganize Result part (see below).

4.2. Results

4.2.1. Comparison of Pre- and Post-Treatment HPLP II across the Three Groups

The pre- and post-test values were analyzed using one-way ANOVA and post hoc test with Sheffé test to determine if significant differences in HPLP II values existed among the three groups. Table 4 reports the means and standard deviations of HPLP II for each group at pre-treatment and post-treatment. Across the six dimensions and overall HPLP II score, it appears that the Pilates group exhibited the most significant improvements, followed by the yoga group and the control group. Participants who participated in the 8-week yoga program had significantly greater improvements in the Nutrition, Physical Activity, and Stress Management subscales than seen in the control group, but were not significantly different from the control in the Interpersonal Relations, Health Responsibility or Spiritual Growth subscales. Volunteers who completed the 8-week Pilates program had significantly greater improvements in all six subscales comparted to yoga and control groups. There were statistically significant differences in post-treatment HPLP II among the three groups, as shown in Table 4. Between-group differences in Interpersonal Relations were statistically significant (F=12.221, p<.001). Between-group differences in Nutrition were also statistically significant (F=40.110, p<.001). Between-group differences in Health Responsibility were statistically significant as well (F=19.383, p<.001). Be-tween-group differences in Health Responsibility were statistically significant (F=35.558, p<.001). Between-group differences in Stress Management were statistically significant (F=30.187, p<.001). Between-group differences in Spiritual Growth were statistically significant (F=19.693, p<.001). Overall, between-group differences in HPLP II at post-treatment were all statistically significant (F=39.446, p<.05).

In our pre-test analyses of HPLP II, when we checked homogeneity in pre-treatment HPLP II across the three groups using one-way ANOVA and Sheffé test, we found statistically significant differences in Interpersonal Relations and Spiritual Growth, two sub-constructs of HPLP II. Accordingly, we needed to check further whether these preexisting differences in Interpersonal Relations and Spiritual Growth values might drive the differences found at post-treatment, using ANCOVA. ANCOVA results are shown in Table 5 and Table 6, which report that there are still significant differences between the three groups for Interpersonal Relations and Spiritual Growth, even after accounting for their pre-treatment values. The ANCOVA result confirms that those initial differences did not change the relationships.

4.2.2. Comparison of Pre- and Post-Treatment HSRS across the Three Groups

Table 7 reports the means and standard deviations of HSRS for each group at pre-treatment and post-treatment. It seems that the Pilates and yoga group also exhibited a considerable increase in HSRS at post-treatment than the control group.

There was no significant difference across the groups when examining ANOVA results at pre-treatment. However, there were statistically significant differences in post-treatment HSRS among the three groups, as shown in Table 7. Between-group differences in Health Status Present were statistically significant (F=11.140, p<.001). Be-tween-group differences in Health Status Compared to One Year Prior were also statistically significant (F=17.522, p<.001). Between-group differences in Health Status Compared to Friends/Colleagues (F=4.915, p<.001) were statistically significant as well. Overall, be-tween-group differences in HSRS at post-treatment were all statistically significant (F=3.720, p<.05).

The pre-test score means were evaluated for preexisting variability between the three groups using one-way ANOVA and Sheffé test. This analysis demonstrated that the three groups did not have significant differences between scores in the three subscales at pre-test (p>0.05).

4.3. Gender Differences

We further analyzed to see if gender-specific differences existed regarding HPLP II and HSRS scores.

Table 8 reports the means and standard deviations of HPLP II for females vs. males at pre- and post-treatment. Gender differences in HPLP II at pre-treatment were not statistically meaningful across all sub-constructs except for Physical Activity. Male participants appear to have reported greater engagement in Physical Activity, compared to female counterparts, at the pre-treatment stage. However, this difference in Physical Activity does not seem to continue after the 8-week exercise program. Although it is worthwhile to note that males appear to have reported slightly larger improvements in Stress Management than females at post-treatment, it is safe to conclude that gender differences in post-treatment changes were not very noticeable in our experimental study as far as the impact of Pilates and yoga on health-promoting behavior is concerned.

Table 9 reports the means and standard deviations of HSRS for females vs. males at pre- and post-treatment, respectively. Gender differences in HSRS at pre-treatment were not statistically meaningful across all sub-constructs except for Health Status Present. Female participants appear to have reported more positive self-perception about their current health, compared to male counterparts, at pre-treatment. However, at post-treatment this difference seems to have disappeared. Again, we do not find much significant gender difference in our experimental study of the impact of Pilates and yoga on other health-promoting behaviors and subjective health status.

Reviewer 3 Report

Although the title informs the variables under study it is necessary to consider the existence of a high variety of research methods. Once the title is of vital importance for the selection and reading of the research reports, it is recommended to identify the report as to the research design used, being consistent with the methodological procedures described. Furthermore, to obtain a more global compression of the subject of study, it is suggested that the authors can specify the "other health-promoting behaviors".

The abstract is organized in a well-structured format. However, it is suggested to introduce the mean and standard deviation of participants in both groups of participants (experimental and control).

The section "Introduction" presents weaknesses regarding the contextualization of the object of study, being too extensive and sometimes redundant in its content. The first paragraph of the "Introduction" section shows redundancies regarding the benefits of physical exercise in health. This paragraph could be synthesized. Furthermore, this report focuses on pilates and yoga practice. As such, it would be preferable that the "Introduction" begins with a contextualization of these physical practices. Also, the way the objective is described does not allow to specify which other health-promoting behaviors the study is intended to study. The report refers to previously published studies to broaden the context and provide an empirical basis for the subsequent development of hypotheses. However, it is suggested that it can be explained how this study aims to address the methodological limitations of previous studies, thus contributing to the justification and relevance of this study. The literature review presented is too extensive. It is suggested to synthesize this section to make the reading more comprehensive for the reader. Furthermore, it is suggested that its rationale be structured to convince the reader of its logic and consistency with another depth. Thus, it would be useful to describe or allude to the premises, concepts, or theoretical models that relate to the object of study. In another dimension, it is suggested that the presentation of the relevance of the main subject understudy to current practical knowledge may present another development, supported by other evidence, previously published, giving it better robustness. The manuscript presents several formatting errors regarding the referencing system that is required by the International Journal of Environmental Research and Public Health: line 76 "well-being Kucuk and Livanelioglu (2015) found". Check throughout the manuscript. Lines 83-98: the information presented is in the form of a "summary" of the study. The presented abstract already reflects the content presented in the paragraph. It is suggested to remove it.

The pilot study lasted 4 weeks. Afterward, the intervention program lasted 8 weeks. The duration of the intervention program and the pilot study were defined based on what theoretical and empirical foundations? It is suggested to clarify. The intervention program is described with a sufficient level of detail. However, tables 1 and 2 can be referred to in the annex. How did the authors ensure that the control group was not inserted in the practice of structured physical exercise? Besides, control group participants may have performed physical activity informally. During the 8 weeks of intervention, were any measures of the control group's level of physical activity and sedentary behavior applied? Clarify. The authors report that an interview was conducted with some of the participants, to allocate them, in a non-random way, to the groups. On which questions were the interview-based? It would be interesting to complement the description of the method with some excerpts from these interviews. It is suggested that the characteristics of the participants of each group can be introduced in the "Results" section. To better understand the universe of the sample, it will be important to clarify which selection and exclusion criteria were defined. Since the sample size plays an important role in the ability to make precise inferences, has any statistical procedure been performed to calculate the sample size? The participants were randomly selected. However, what was the random sampling strategy used? The "Method" section presents a serious failure in not presenting the statistical procedures used.

The presentation of results in textual form is not perceptible to the reader. presenting a high number of tables. Moreover, the figures and tables show duplicate data in the tables and text. It is suggested to reformulate the presentation of results. The tables presented have different formatting. This should be corrected.

The "Discussion" section could present a more comprehensive development and relate the results of this research with the previously published studies, referenced in the "Introduction", related to the subject of study. Besides, explanations based on robust theoretical models could be provided. It is not clear the possible reasons why the practice of pilates presents a greater advantage compared to the practice of yoga. Besides, it is necessary to highlight the methodological limitations of this study, suggesting other procedures to overcome them. The conclusion of the study is not perceptible. It is recommended that, instead of presenting a literature analysis and a summary of the report, a general, clear interpretation of the results evidenced here be carried out.

Author Response

Response to Reviewer 3

We have revised the manuscript in response to the helpful and detailed comments you provided. In our view, this revision has led to a great deal of improvement, further strengthening our paper. We are very grateful for your feedback that has led to these improvements. In the detailed response below, we describe our efforts to respond to the concerns and questions raised. We hope you will find the new manuscript clearer and of better quality.

Below are our point-by-point responses to each of your comments.

Title

1. Although the title informs the variables under study it is necessary to consider the existence of a high variety of research methods. Once the title is of vital importance for the selection and reading of the research reports, it is recommended to identify the report as to the research design used, being consistent with the methodological procedures described. Furthermore, to obtain a more global compression of the subject of study, it is suggested that the authors can specify the "other health-promoting behaviors".

Given that Pilates and yoga are also a subset of health-promoting behaviors, we initially thought that by adding ‘other’ we could emphasize a kind of spillover influence of Pilates and Yoga on ‘other’ health-promoting behaviors.

But on the second thought, we now think that adding “other” could cause unnecessary confusion for readers. So the title is corrected as follows. And 'other' was deleted from the main part as well.

We appreciate your clarifying question.

The Impacts of Pilates and Yoga on Health-Promoting Behaviors and Subjective Health Status

Abstract

1. The abstract is organized in a well-structured format. However, it is suggested to introduce the mean and standard deviation of participants in both groups of participants (experimental and control).

Based on your suggestion, we have revised our Abstract and included the information about the subjects’ age, sex composition, and tests deployed (yellow highlighted) in this Abstract as below.

Abstract: This study investigates whether Pilates and yoga lead people to adopt generally health-promoting lifestyle elements and feel better about their physical and mental fitness. To this end, we designed an 8-week exercise program of Pilates and yoga reviewed by veteran practitioners and conducted an experimental study through which we collected the data from 90 volunteered adult subjects between ages 30 and 49 (mean age=35.47), equally represented by women and men without previous experience with Pilates or yoga. In the 8-week long experiment, we assigned the subjects to three groups, where subjects in the two exercise groups regularly took part in either Pilates or yoga classes, and the control group participated in neither exercise classes. All participants completed two surveys, the Health-Promoting Lifestyle Profile (HPLP II) and the Health Self-Rating Scale (HSRS), before and after their assigned program. In our analysis of pre- and post-treatment differences across the three groups, we ran ANOVA, ANCOVA, and Sheffe’s test implemented using SPSS PASW Statistics 18.00. Our results indicate that Pilates and yoga groups exhibited a higher engagement in health-promoting behaviors than the control group after the program. Subjective health status, measured with HSRS, also improved significantly among Pilates and yoga participants compared to those in the control group after the program. The supplementary analysis finds no significant gender-based difference in these impacts. Overall, ….

Introduction

1. The section "Introduction" presents weaknesses regarding the contextualization of the object of study, being too extensive and sometimes redundant in its content. The first paragraph of the "Introduction" section shows redundancies regarding the benefits of physical exercise in health. This paragraph could be synthesized. Furthermore, this report focuses on pilates and yoga practice. As such, it would be preferable that the "Introduction" begins with a contextualization of these physical practices. Also, the way the objective is described does not allow to specify which other health-promoting behaviors the study is intended to study. The report refers to previously published studies to broaden the context and provide an empirical basis for the subsequent development of hypotheses. However, it is suggested that it can be explained how this study aims to address the methodological limitations of previous studies, thus contributing to the justification and relevance of this study. The literature review presented is too extensive. It is suggested to synthesize this section to make the reading more comprehensive for the reader. Furthermore, it is suggested that its rationale be structured to convince the reader of its logic and consistency with another depth. Thus, it would be useful to describe or allude to the premises, concepts, or theoretical models that relate to the object of study. In another dimension, it is suggested that the presentation of the relevance of the main subject understudy to current practical knowledge may present another development, supported by other evidence, previously published, giving it better robustness.

Thank you for these comments and suggestions. We have rewritten and reorganized this part (see below).

2. The manuscript presents several formatting errors regarding the referencing system that is required by the International Journal of Environmental Research and Public Health: line 76 "well-being Kucuk and Livanelioglu (2015) found". Check throughout the manuscript.

Thank you for your comments. We have rewritten and reorganized this part (see below, highlighted in yellow).

3. Lines 83-98: the information presented is in the form of a "summary" of the study. The presented abstract already reflects the content presented in the paragraph. It is suggested to remove it.

Thank you for these comments and suggestions. We have rewritten and reorganized this part (see below).

  1. Introduction

Less known, however, is whether Pilates and yoga may also have secondary, indirect benefits. Adopting a broad set of lifestyle elements that are generally regarded as beneficial to one’s physical and psychological health and holding a positive belief about one’s health might be necessary for leading a healthy life. Some prior research hints at the possibility that one form of health-promoting behavior triggers people to adopt broadly health-promoting lifestyle elements (e.g., healthy eating, enriching interpersonal relationships, informed health-related decisions, work-life balance, emphasis on spiritual growth, etc.), thereby setting in motion a positive reinforcement loop. For example, Kucuk and Livanelioglu (2015) found that individuals starting a clinical Pilates program tend to hold a positive notion about the goodness of exercises and experience an improvement in their socio-psychological aspects of lives [11]. Mustian’s research team (2013) found that yoga has a positive impact on the sleep quality among cancer survivors [13]. Galasso and colleagues (2020) found that aerobic and anaerobic exercise training was effective in reducing the tendency of binge eating [14].

However, these studies come short of providing more comprehensive and clear evidence that Pilates leads to a positive change in both the participants’ overall health-promoting behavior and self-perceived health status. Besides, Kucuk and Livanelioglu (2015)’s study has primarily targeted people taking a clinical Pilates program, excluding those who might regularly engage in Pilates for non-clinical purposes [11]. And Pilates was not covered in studies by Mustian et al. (2013) and Galasso et al. (2020) [13, 14]. Given this, the current study seeks to fill this deficiency in prior work by directly investigating whether regularly engaging in an exercise program such as Pilates and yoga might positively affect participants’ adoption of health-promoting behaviors. Additionally, we examine the extent to which these outcomes shown in program participants are accompanied by an increase in self-perceived physical and mental well-being.

Methods

1. The pilot study lasted 4 weeks. Afterward, the intervention program lasted 8 weeks. The duration of the intervention program and the pilot study were defined based on what theoretical and empirical foundations? It is suggested to clarify.

Thank you for your comments. We provided a more clear explanation for the pilot test (see below).

3.2.1. Pilot Test

In November 2017, a pilot test was started with nine people who had participated in a 4-week exercise program. Our 4-week pilot study aimed to ensure the appropriate design for our main experiment to be followed. Our focus was to configure the exercise program's adequate task complexity levels in advance of the full-scale experiment to last over eight weeks.

The pilot test subjects were informed that they were participating in some study on a community health program and that their personal information would be kept confidential. Participants were randomly assigned to each of the three groups: a 4-week Pilates program, a 4-week yoga program, or the control group with no specified exercise program.

During the pilot study, we noticed that some of the exercises we initially included in the program were quite challenging and problematic for some people. For example, the roll-over movement in Pilates turned out to be rather hard for those with no prior exposure to Pilates and yoga. It was crucial to ensure that the exercises included in each program were safe and secure, and suitable for beginning and intermediate-level students who do not have any Pilates or yoga experience. Given that people usually have a weak core and unstable lumbopelvic stability, the rolling-over, inversion movements, or extreme stretching were removed to minimize any injury or accident during the sessions.

2. The intervention program is described with a sufficient level of detail. However, tables 1 and 2 can be referred to in the annex.

Thank you for your suggestions.

Following your advice, we provided these tables in Appendix A.

3. How did the authors ensure that the control group was not inserted in the practice of structured physical exercise? Besides, control group participants may have performed physical activity informally. During the 8 weeks of intervention, were any measures of the control group's level of physical activity and sedentary behavior applied? Clarify.

The authors report that an interview was conducted with some of the participants, to allocate them, in a non-random way, to the groups. On which questions were the interview-based? It would be interesting to complement the description of the method with some excerpts from these interviews.

Thank you for your comments. We clarified in 3.3 Participants section as below.

3.3. Participants

Our inclusion criteria were age 30-49 years, no severe disease, and have interest in yoga and Pilates. Our exclusion criteria were that having prior experience for yoga and Pilates and currently participating regularly in exercises such as swimming and regular fitness work-outs.

Yoga Community and Korea Pilates Federation assisted recruitment of participants and instructors. In November 2017, the first author contacted and visited the national Pilates Federation and several local yoga communities across South Korea. In meetings with directors of these organizations, the first author explained the purpose of this study. The directors and organizations who agreed to participate in this study assisted in recruiting volunteer-participants. Potential participants were identified through these contacts and inquired about their interest and willingness to participate in this study. A sufficient number of people (approximately 100) expressed their desire to participate in our research.

Two exercise groups of 30 individuals were randomly selected with a balanced gender ratio for each group. The Pilates and yoga group participated in an 8-week pre-designed exercise program.

We also needed the control group who would not receive the treatment (i.e., no exercise) but would take the two surveys following their instructors' directions. The control group did not participate in any of the specified exercise programs to serve as a baseline comparison for the experiment. Individuals in the control group comprised those who expressed interest in taking a Pilates or yoga class but could not make it for scheduling conflicts. However, they agreed to complete the two surveys at the same 8-week interval as the Pilates and yoga groups. Throughout the 8-week study, we ensured that they did not engage in any other exercise programs during the study period. 

Results

1. It is suggested that the characteristics of the participants of each group can be introduced in the "Results" section. To better understand the universe of the sample, it will be important to clarify which selection and exclusion criteria were defined. Since the sample size plays an important role in the ability to make precise inferences, has any statistical procedure been performed to calculate the sample size? The participants were randomly selected. However, what was the random sampling strategy used? The "Method" section presents a serious failure in not presenting the statistical procedures used.

Thank you for your suggestions. Following your advice, we did revise Results section. 

2. The presentation of results in textual form is not perceptible to the reader. presenting a high number of tables. Moreover, the figures and tables show duplicate data in the tables and text. It is suggested to reformulate the presentation of results. The tables presented have different formatting. This should be corrected.

Thank you for your suggestions. Following your advice, we did revise Results section. 

Discussion

1. The "Discussion" section could present a more comprehensive development and relate the results of this research with the previously published studies, referenced in the "Introduction", related to the subject of study. Besides, explanations based on robust theoretical models could be provided. It is not clear the possible reasons why the practice of pilates presents a greater advantage compared to the practice of yoga. Besides, it is necessary to highlight the methodological limitations of this study, suggesting other procedures to overcome them.

Thank you for your comments.

Following your advice, we revised our Discussion. (see below)

  1. Discussion

The idea that exercise is essential for maintaining a healthy life is almost a truism these days. However, why do some people succeed in adopting a set of health-promoting lifestyle elements, while other people keep failing to stay away from lifestyle elements detrimental to their health? In other words, why do some of them come short of maintaining a healthy lifestyle long enough despite their initial intention to lead a healthy life? Given the importance of this question, understanding the mechanism by which people set in motion the virtual cycle of leading a healthy life is crucial among health care professionals and the general public alike. Indeed, it has become increasingly crucial in public health care research to have a more nuanced understanding of how different aspects of people's lives affect their overall well-being and the role of physical activity in this process.

In this study, we have examined whether Pilates and yoga participation triggers health-promoting behaviors and positively influences on self-evaluation of health status among the participants. Specifically, we have sought to address this question by establishing a causal relationship, not merely a correlation, by comparing the participants’ self-reported changes in health-promoting behaviors—including eating healthy, avoiding a sedentary lifestyle, being responsible for own health, maintaining healthy social relationships, managing stress, and emphasizing spiritual growth—and health status —compared to the prior year and peers—before and after the pre-designed 8-week exercise program.

In our analysis of the data collected from 90 female and male adult volunteers, we found that the Pilates group reported the most considerable improvements in the two measures of HPLP II and HSRS, each of which corresponds to self-assessed health-promoting behavior and health status. The increases at post-treatment for the Pilates group appeared even larger than those reported by the yoga group participants. This finding implies that Pilates offers a greater advantage over yoga as an exercise method.

The result of this study revealed that engagement in Pilates or yoga for eight weeks did result in improved rates of reported engagement in health promoting behaviors relative to a control group using the HPLP II assessment tool. While improvements in the Physical Activity subscale were direct and expected, improvements were also seen in the Nutrition and Stress Management subscales for Pilates and yoga and Interpersonal Relations, Health Responsibility, and Spiritual Growth subscales for Pilates to the control.

Other than Stress Management, all other HPLP II subscales did not show significant differences in outcomes for males and females. Also, HSRS scores showed no significant differences in outcomes between male and female groups. This indicates that the benefits seen from doing Pilates and yoga are not gender-specific, and thus promotion of these exercise programs should not be gender restricted. Given that we found no considerable gender differences in the health- promoting impacts of Pilates and yoga, Pilates and yoga can be an effective public health program among both men and women when appropriately implemented.

Despite some similarities, Pilates and yoga do have some notable differences. Here, we briefly discuss some notable differences between Pilates and yoga. First, as depicted in the earlier section, Pilates and yoga have different historical origins, which appear to influence how they have been evolved and organized as a system of fitness program. Pilates is a relatively new invention that started in the early 20th century, while yoga began more than 5,000 years ago. Pilates has been successfully evolved as a standardized fitness program and organized as a profession due to its relatively contemporary nature. The formation of the PMA (Pilates Method Association) in 2001, a non-profit professional association, has accelerated the standardization process by clearly delineating the scope of practice. PMA has also transformed many trainers and instructors into an organized profession by administrating a centralized system of qualification and setting professional standards, and establishing the code of ethics in interactions with clients.

In contrast, yoga has variegated and decentralized into numerous branches, which seems to have slowed any standardized and centrally organized attempt. Second, while both Pilates and yoga emphasize harmonization among mind, body, and spirit, their exercise approaches are somewhat different. For example, the breathing techniques utilized differ between Pilates and yoga. Moreover, whereas Pilates tends to focus more on effective muscle energy movement using dynamic, resistance, and stability-related poses, yoga focuses more on how a person feels through relaxation, meditation, and static stretching. Accordingly, Pilates exercises could yield better results more quickly. However, our study has limitations in revealing the mechanisms behind the varied health-related benefits of Pilates and yoga. This topic merits a more detailed investigation in future research with a longer-term intervention.

Our findings are similar to some prior studies, albeit with differences in targeted subjects, that have found associations between engaging in Pilates and yoga exercise programs and perceived health. For instance, in a study by Küçük and Livanelioglu (2015), participating in Pilates led to improvements in healthy women's perception of exercise and other psychosocial variables [11]. In the study of Berent et al. (2014), yoga was demonstrated to be beneficial for promoting improvements in college students' quality of life and healthy lifestyle choices [45]. Neumark-Sztainer et al. (2010) examined the associations between participating in mind-body activities such as Pilates and yoga and the prevalence of body dissatisfaction and eating disorders in young adults [46]. In their study, most individuals with body dissatisfaction and eating disorder behaviors were reduced by participation in yoga and Pilates.

However, our findings depart from previous research highlighting Pilates and yoga's effects on body composition changes and weight loss. In this study, we found only modest decreases in BMI among the Pilates and yoga program participants. This is probably attributed to the fact that our participants’ BMI values were within the average range even before participating in the exercise program. Additional research incorporating participants with obesity would help determine the comparative effectiveness of Pilates vs. yoga on body composition. Given that Pilates focuses more on core control and posture development. In contrast, yoga focuses more on static stretching and flexibility, it would also be interesting to see how their effects on body composition might differ among subjects with obesity. Follow-up research can also further examine the health-promoting benefits of Pilates and yoga program on people with acute or chronic illnesses and its linkage to their rehabilitation.

Given that most people find it challenging to maintain a healthy lifestyle long enough, contrary to their desire to do otherwise, our study provides valuable insights into how to set in motion a virtuous loop of a healthy life. People who practice Pilates or yoga do so not just to increase the amount of their physical activity but also to restore a balance in their hectic everyday lives. This paper has also illuminated one avenue through which Pilates and yoga can help ordinary men and women restore a healthy balance in life. Overall, our findings have useful implications for health policy-makers and authorities by offering more clear evidence of Pilates and yoga's indirect benefits.

This study has several limitations. The study participants were all ethnic Korean residing in South Korea, so that the results may differ for other ethnicities or locations. This study's participants were limited to ages 30 to 49, constraining the generalizing our findings beyond this group. Also, subsequent studies are needed to test how different mixtures of exercise programs in Pilates and yoga might affect health-promoting behavior.

Conclusion

1. The conclusion of the study is not perceptible. It is recommended that, instead of presenting a literature analysis and a summary of the report, a general, clear interpretation of the results evidenced here be carried out.

Thank you for your comments.

Following your advice, we revised this section trying to be more concise in our description.

(see below)

  1. Conclusions

From a public health standpoint, it is essential to ensure that people practice a healthy lifestyle with a belief about its positive effect on their health. Such a lifestyle should include a wholistic approach, taking both physical and psychological aspects of well-being seriously [12, 46-49]. In this article, we have extended prior research suggesting that various elements of health-promoting behaviors can reinforce one another, high-lighting and demonstrating the triggering role of exercise such as Pilates and yoga [12, 46-49]. Overall, our results confirm that Pilates and yoga help recruit health-promoting behaviors in participants and engender positive beliefs about their subjective health status, thereby setting a positive reinforcement cycle in motion. By providing clear evidence that the promotion of Pilates or yoga can serve as an effective intervention strategy that helps individuals change behaviors adverse to their health, this study offers practical implications for health care professionals and public health officials alike. Our work has shed further light on the relationship between participation in such low-risk and readily accessible exercise programs as Pilates and yoga and the likelihood that people engage in comprehensive practices of promoting their health.  

Round 2

Reviewer 2 Report

I appreciate the changes that have been made, which allow us to give the study a higher quality

Author Response

Thank you for your kind remarks. We really appreciate it.

Reviewer 3 Report

It appears that the authors have made most of the changes requested in the previous review report. However, there are still aspects that need further explanation.  Also, we would like to add the following comments:

The pilot study lasted 4 weeks. Afterward, the intervention program lasted 8 weeks. The duration of the intervention program and the pilot study were defined based on what theoretical and empirical foundations? It is suggested to clarify.

How did the authors ensure that the control group was not inserted in the practice of structured physical exercise? Besides, control group participants may have performed physical activity informally. During the 8 weeks of intervention, were any measures of the control group's level of physical activity and sedentary behavior applied? Clarify.

In the first version of the manuscript, the authors report that an interview was conducted with some of the participants, to allocate them, in a non-random way, to the groups. On which questions were the interview-based?

The second version of the article presents the limitations of the study. However, the limitations presented focus only on the characteristics of the participants. It is suggested that other methodological limitations should be presented. For example, there are moderating and mediating variables that were not analyzed and that may introduce biases in the results presented.

It is also suggested that concrete practical implications be presented based on the results presented. The implications can be based on the training program presented.

Author Response

Response to Reviewer 3

Thank you very much for your clarifying questions and comments. We should have made it more explicit in our previous round of revision in tackling the questions and concerns you raised. In this letter, we will do our best to explain how we have addressed your questions and concerns about our study. We have also further revised our manuscript to incorporate your suggestions in this 2nd round of revision. We believe that our manuscript has improved considerably through these two revisions, and so we are very grateful for your time and efforts in reviewing our paper. 

Below are our point-by-point responses to each of your comments.

1.     The pilot study lasted 4 weeks. Afterward, the intervention program lasted 8 weeks. The duration of the intervention program and the pilot study were defined based on what theoretical and empirical foundations? It is suggested to clarify.

Our decision for the duration of the intervention program and the pilot study—eight weeks and four weeks, respectively—are based on the foundational work of Joseph Pilates along with several follow-up empirical studies.

Our pilot experiment was performed three times a week for four weeks, adding up to 12 lessons. The lessons in our main experiment were held 24 times for eight weeks.

According to Joseph Pilates, subjects could see some change after ten times, major change after 20 lessons, and considerable change after 30 lessons. We also assumed this principal can be applied to yoga.

In this light, our study design is consistent with prior studies on the effects of Pilates and yoga. We list these studies:

8-week Pilates Clinical Pilates is effective in changing exercise beliefs and physical and psychosocial parameters.

Küçük, F., & Livanelioglu, A. (2015). Impact of the clinical Pilates exercises and verbal education on exercise beliefs and psychosocial factors in healthy women. Journal of Physical Therapy Science, 27(11), 3437–3443.

6-week Pilates-based exercises, performed along with standard exercise programs, were found to be effective in improving the balance and quality of life.

Karaman A, Yuksel I, Kinikli GI, Caglar O. (2017) Do Pilates-based exercises following total knee arthroplasty improve postural control and quality of life? Physiotherapy Theory and Practice, 33(4), 289-295.

8-week yoga training program was effective in reducing the severity of symptoms in children with autism.

Sotoodeh, MS., Arabameri, E., Panahibakhsh, M., Kheiroddin, F., Mirdoozandeh, H., Ghaizadeh, A., (2017). Effectiveness of yoga training program on the severity of autism. Complementary Therapies in Clinical Practice, 28, 47-53.

As we noted in the manuscript, a pilot experiment was conducted to verify appropriate pre- and post-test task complexity levels, examine the proper exercise program and minimize differences among the three groups before the full-scale experiment. Our pilot participants did not participate in the main experiment.

During the one-month pilot program, we learned that our initial exercise standard was quite challenging and some movements could cause some safety concerns for some participants. For example, roll-over exercise was hard to perform for beginners with no prior Pilates experience. Therefore, we decided to exclude the rolling over or inversion exercises in the main experiment.

2.     How did the authors ensure that the control group was not inserted in the practice of structured physical exercise? Besides, control group participants may have performed physical activity informally. During the 8 weeks of intervention, were any measures of the control group's level of physical activity and sedentary behavior applied? Clarify.

We ensured that our control group maintains their pre-test lifestyles and activity levels, similar to prior studies in their investigation of Pilates and yoga effects. At the beginning of the experiment, the control group subjects also signed the consent form to follow the instructions. They were instructed to maintain their lifestyle and activity levels constant from the start until the end of the experimental period with no insertion of new physical exercises. Further, we had our instructors check in with them once a week via the messenger program to see whether they were maintaining to refrain from beginning a new exercise program or changing their current activity levels.

Pre- and post-test of these subjects' body measures reveal that there was not much difference during the experiment, as shown in the table below. 

Control (n=30)

Pre-test

Mean ± SD

Post-test

Mean ± SD

Waist circumference(inch)

29.40±3.024

29.60±3.500

Hip circumference(inch)

34.53±2.623

34.47±2.763

BMI

22.09±3.050

22.39±3.077

We list previous work employing a similar approach as below:

Irez, G.B.; Ozdemir, R.A; Evin R.; Irez, S.G.; Korkusuz, F. Integrating Pilates Exercise into an Exercise program for 65+ year-old Women to reduce Falls. J Sports Sci Med. 2011, 10, 105-111.

Natour, J.; Cazotti, L. D.; Ribeiro, L. H.; Baptista, A. S.; Jones, A. Pilates Improves Pain, Function and Quality of life in Patients with Chronic Low Back Pain: A randomized controlled trial. Clin Rehabil. 2014, 29, 59-68.

Hewett, Z. L.; Pumpa, K. L.; Smith, C. A.; Fahey, P. P.; Cheema, B. S. Effect of a 16-week Bikram Yoga Program on Perceived stress, Self-efficacy and Health-related Quality of life in Stressed and Sedentary Adults: A randomized controlled trial. J Sports Sci Med. 2017, 21, 352-357.

3.     In the first version of the manuscript, the authors report that an interview was conducted with some of the participants, to allocate them, in a non-random way, to the groups. On which questions were the interview-based?

We conducted interviews with respondents who said yes to the questions about their current exercise status to determine the details about these exercises further. We decided to exclude the 6 participants who were engaging in a fitness program, swimming, and extreme physical activity based on these interviews. Below is a part related to your question in our manuscript.

  3.3.1. Pre-screening

A total of 90 volunteers participated in this project. All participants took their first Pilates or yoga class or did not have prior experience with Pilates or yoga. We provided them with a privacy statement stipulating that their personal information would remain strictly confidential.

Before starting the exercise program designed, we screened pre-treatment survey responses of our volunteer-participants regarding their characteristics and exercise status to ensure that they are similar in those aspects. Through the survey and on-site measurement equipment, we obtained basic demographic information (gender, age, education level, occupation) and anthropometric data (height, weight, waist circumference, hip circumference, body mass index (BMI)). We measured the participants’ BMI using In-Body equipment at studio (e.g., InBody Dial W version 2.3.05, Korea).

Of high importance was to ask the participants whether they had been involved in any other exercise programs. This question was necessary to mitigate against any systematic bias their prior or recent exposure to exercise programs of any kind can introduce. Later, we checked whether there is any systematic difference between people currently practicing other exercises and those not engaging in any other exercise program. These pilot test participants did not take part in the main experiment.

In the survey conducted at the pre-test, a subject is to answer the three questions about her exercise status: (1) I do exercise for health; (2) I do regular exercise; (3) I know a proper exercise program. To ensure homogeneity of different exercise status between the three groups, we ran a Chi-square test to see whether there are differences between the three groups in exercise status.

We ran the homogeneity test to see participants were equivalent. In particular, it was critical to check their exercise status because participants’ exercise status can serve as a potential interference in our experiment due to its expected influence on HPLP II and HSRS. People who exercise for health and know of a proper exercise method turned out to be concentrated in the control group at a statistically meaningful level (X2 =10.075, p<.01, X2=8.285, p<.05). This means that people who exercise for health and know of a proper exercise method are not randomly assigned to each group. Accordingly, the first author underwent an interview with those who answered positively to these questions to find out more about their exercise history. Then, six volunteers assigned to the control group with notable exercise history were replaced with other volunteers without such exercise history before our main experiment.

4.     The second version of the article presents the limitations of the study. However, the limitations presented focus only on the characteristics of the participants. It is suggested that other methodological limitations should be presented. For example, there are moderating and mediating variables that were not analyzed and that may introduce biases in the results presented.

Thank you for your suggestion. Following your advice, we have added further limitations related to methodological issues that were not directedly covered in our study.

This study has a few limitations. The study participants were all ethnic Korean re-siding in South Korea, so that the results may differ for other ethnicities or locations. This study's participants were limited to ages 30 to 49, constraining the generalizing of our findings beyond this group. Also, subsequent studies are needed to test how different mixtures of exercise programs in Pilates and yoga might affect health-promoting behavior. Future studies are needed to shed further light on the mechanisms behind the differences between Pilates and yoga. For example, what might be intervening variables in yielding differences in the effects of Pilates and yoga? Given that these two exercises have a slightly different focus, researchers can further examine this question by measuring other factors related to subjects’ self-efficiency and self-control and differences in seeking the balance between physical and mental aspects in life (610-620).

5.     It is also suggested that concrete practical implications be presented based on the results presented. The implications can be based on the training program presented.

Thank you for your suggestion. Following your advice, we have added concrete practical implications based on our results presented as follows:

In this study, Pilates and yoga program were designed by a master instructor who was certified with an international license and had over ten years of experience. Those exercises, including goals and precautions, were safe for all people and did not include any inversion exercise or extreme stretching to avoid spine and pelvis issues. Therefore, the implemented program of Pilates and yoga can be applied to the general public (604-609).
